# An in vitro model of tumor heterogeneity resolves genetic, epigenetic, and stochastic sources of cell state variability

Corey E. Hayford[1], Darren R. Tyson[2], C. Jack Robbins, III[2¤a], Peter L. Frick[1¤b], Vito Quaranta[2,3], Leonard A. Harris[4,5,6]*

1 Chemical and Physical Biology Graduate Program, Vanderbilt University School of Medicine, Nashville, Tennessee, United States of America, 2 Department of Biochemistry, Vanderbilt University School of Medicine, Nashville, Tennessee, United States of America, 3 Department of Pharmacology, Vanderbilt University School of Medicine, Nashville, Tennessee, United States of America, 4 Department of Biomedical Engineering, University of Arkansas, Fayetteville, Arkansas, United States of America, 5 Interdisciplinary Graduate Program in Cell and Molecular Biology, University of Arkansas, Fayetteville, Arkansas, United States of America, 6 Cancer Biology Program, Winthrop P. Rockefeller Cancer Institute, University of Arkansas for Medical Sciences, Little Rock, Arkansas, United States of America

¤a Current address: MD-PhD Graduate Program, Yale University School of Medicine, New Haven, Connecticut, United States of America
¤b Current address: Intuit Inc., Mountain View, California, United States of America
* harrisl@uark.edu

**Data Availability Statement:** The sequencing datasets generated in this study can be found in the gene expression omnibus (GEO; GSE150084) and sequence read archive (SRA; PRJNA631050 and

## Abstract

Tumor heterogeneity is a primary cause of treatment failure and acquired resistance in cancer patients. Even in cancers driven by a single mutated oncogene, variability in response to targeted therapies is well known. The existence of additional genomic alterations among tumor cells can only partially explain this variability. As such, nongenetic factors are increasingly seen as critical contributors to tumor relapse and acquired resistance in cancer. Here, we show that both genetic and nongenetic factors contribute to targeted drug response variability in an experimental model of tumor heterogeneity. We observe significant variability to epidermal growth factor receptor (EGFR) inhibition among and within multiple versions and clonal sublines of PC9, a commonly used EGFR mutant nonsmall cell lung cancer (NSCLC) cell line. We resolve genetic, epigenetic, and stochastic components of this variability using a theoretical framework in which distinct genetic states give rise to multiple epigenetic "basins of attraction," across which cells can transition driven by stochastic noise. Using mutational impact analysis, single-cell differential gene expression, and correlations among Gene Ontology (GO) terms to connect genomics to transcriptomics, we establish a baseline for genetic differences driving drug response variability among PC9 cell line versions. Applying the same approach to clonal sublines, we conclude that drug response variability in all but one of the sublines is due to epigenetic differences; in the other, it is due to genetic alterations. Finally, using a clonal drug response assay together with stochastic simulations, we attribute subclonal drug response variability within sublines to stochastic cell fate decisions and confirm that one subline likely contains genetic resistance mutations that emerged in the absence of drug treatment.

PRJNA632351). Additional experimental data are available on Github (github.com/QuLab-VU/GES_2021). The codes used to generate model simulations and analyze experimental data are publicly available via GitHub (github.com/QuLab-VU/GES_2021).

**Funding:** This work was supported by the following funding sources: C.E.H., National Institutes of Health (NIH) Ruth L. Kirschstein National Research Service Award (NRSA, F31-CA221147) and Chemical-Biology Interface Training Grant (T32-GM0650); L.A.H., Vanderbilt Biomedical Informatics Training Program (NLM 5T15-LM007450-14), Quantitative Systems Biology Center at Vanderbilt, and National Cancer Institute (NCI) Transition Career Development Award to Promote Diversity (K22-CA237857-01A1); D.R.T., Lung Cancer Research Foundation (LCRF, UALC 13020513) and NIH Research Specialist Award (1R50CA243783); P.L.F., NIH NRSA (F31-CA165840); C.J.R., Vanderbilt Trans-Institutional Programs Grant: Understanding the Complexity of Life One Cell at a Time; V.Q., NIH Clinical and Translational Science Award (U54-CA113007); Sequencing studies were supported by the Vanderbilt Institute for Clinical and Translational Research (VICTR, Voucher VR52385). The funders had no role in study design, data collection and analysis, decision to publish, or preparation of the manuscript.

**Competing interests:** The authors have declared that no competing interests exist.

**Abbreviations:** BP, Biological Process; CC, Cellular Component; cFP, clonal fractional proliferation; CNV, copy number variant; CSC, cancer stem cell; CV, coefficient of variation; DEG, differentially expressed gene; DIP, drug-induced proliferation; DMSO, dimethyl sulfoxide; EGFR, epidermal growth factor receptor; EGFRi, EGFR inhibitor; FACS, fluorescence-activated cell sorting; GATK, Genome Analysis Toolkit; GHR, Genetics Home Reference; GO, Gene Ontology; GTF, gene transfer format; HTO, hashtag oligonucleotide; InDel, insertion/deletion; MF, Molecular Function; MSigDB, molecular signatures database; NSCLC, nonsmall cell lung cancer; PCA, principal component analysis; scRNA-seq, single-cell RNA sequencing; SNP, single nucleotide polymorphism; SSA, stochastic simulation algorithm; t-SNE, t-distributed Stochastic Neighbor Embedding; UMAP, Uniform Manifold Approximation and Projection; UMI, unique molecular identifier; VEP, Variant Effect Predictor; WES, whole exome sequencing.

## Introduction

Cancer is a complex and dynamic disease characterized by intertumoral and intratumoral heterogeneities that have been implicated in treatment avoidance and acquired resistance to therapy [1,2]. Genetic differences among cancer cells within and across tumors have long been appreciated [3–8]. Indeed, genomic instability is a hallmark of cancer [9,10] and is considered to be the primary source of this genetic heterogeneity [11,12]. However, it is becoming increasingly apparent that genetics alone cannot fully explain the wide ranges of responses observed in patient populations to anticancer therapies [13,14]. Epidermal growth factor receptor (EGFR) inhibitors, for instance, are not equally effective across EGFR mutant lung cancer patients, and in almost all cases, tumors eventually acquire resistance and recur [15,16]. Researchers are, therefore, increasingly looking to nongenetic sources of tumor heterogeneity for explanations. These include factors such as cell type of origin, microenvironmental differences between primary and metastatic sites, spatial variations in the microenvironment of an individual tumor, cell plasticity, cell–cell interactions, probabilistic cell fate decisions, and noise in gene expression [17]. Broadly speaking, nongenetic heterogeneity comes in 2 forms [13,18–22]: epigenetic, which is heritable [18,20] (for at least a few generations), and stochastic, which is not heritable and arises due to intrinsic or extrinsic factors, such as gene expression noise [23–27], asymmetric cell division [28,29], or environmental fluctuations. Nongenetic heterogeneity has been linked to drug tolerance and decreased drug sensitivity in vitro [2,30–33], in vivo [30,31,34], and clinically [35,36].

A theoretical concept that connects genetic and nongenetic sources of tumor heterogeneity is the "epigenetic landscape," proposed by Waddington over 50 years ago [37] but has received renewed attention recently [19,20,38,39] (see Table A in S1 Text for definitions of terms useful for the following discussion). In analogy to the potential energy landscape of physical chemistry [40], Waddington posited that the state of a cell can be assigned a "quasi-potential energy" and placed within a landscape where basins (local minima) correspond to cellular phenotypes. Phenotypic state transitions occur when cells traverse the barriers separating adjacent basins, driven by intrinsic (e.g., gene expression) or extrinsic sources of noise [41,42]. At a fundamental level, the state of a cell arises from the complex set of biochemical interactions within (and possibly across [17,43,44]) cells that drive cellular behavior [45,46]. The rates at which these interactions occur depend strongly on protein structure (e.g., the accessibility of a binding domain), which can be changed by so-called "activating" mutations [47]. Thus, one can think of an epigenetic landscape as deriving from a given genetic state and genetic mutations as altering that landscape [19,39] (Fig 1; see S1 Text for further discussion). Typical tumors comprise numerous genetic states [6,48] and are thus expected to harbor numerous overlapping epigenetic landscapes, each of which is subject to noise-induced phenotypic transitions. Changes in environmental factors and treatment with pharmacological agents can also alter these landscapes [33]. Note that the epigenetic landscape was originally devised as a way to explain cellular differentiation during development. In fact, a developmental hierarchy is a special type of epigenetic landscape, where successive basins decrease in quasi-potential energy, leading cells to descend from one state to the next. In cancer, rather than a well-defined hierarchy, multiple basins of comparable depth coexist. A population of cancer cells is thus expected to spread out across these basins, resulting in a highly heterogeneous population [49]. This may confer a survival benefit to the population in the face of future stressors, such as drug treatments [33,50].

Here, we utilize this three-tiered view of tumor heterogeneity (Fig 1) to resolve genetic, epigenetic, and stochastic sources of variability within a family of cell line "versions" and clonal sublines of PC9, a commonly used nonsmall cell lung cancer (NSCLC) cell line. Based on the

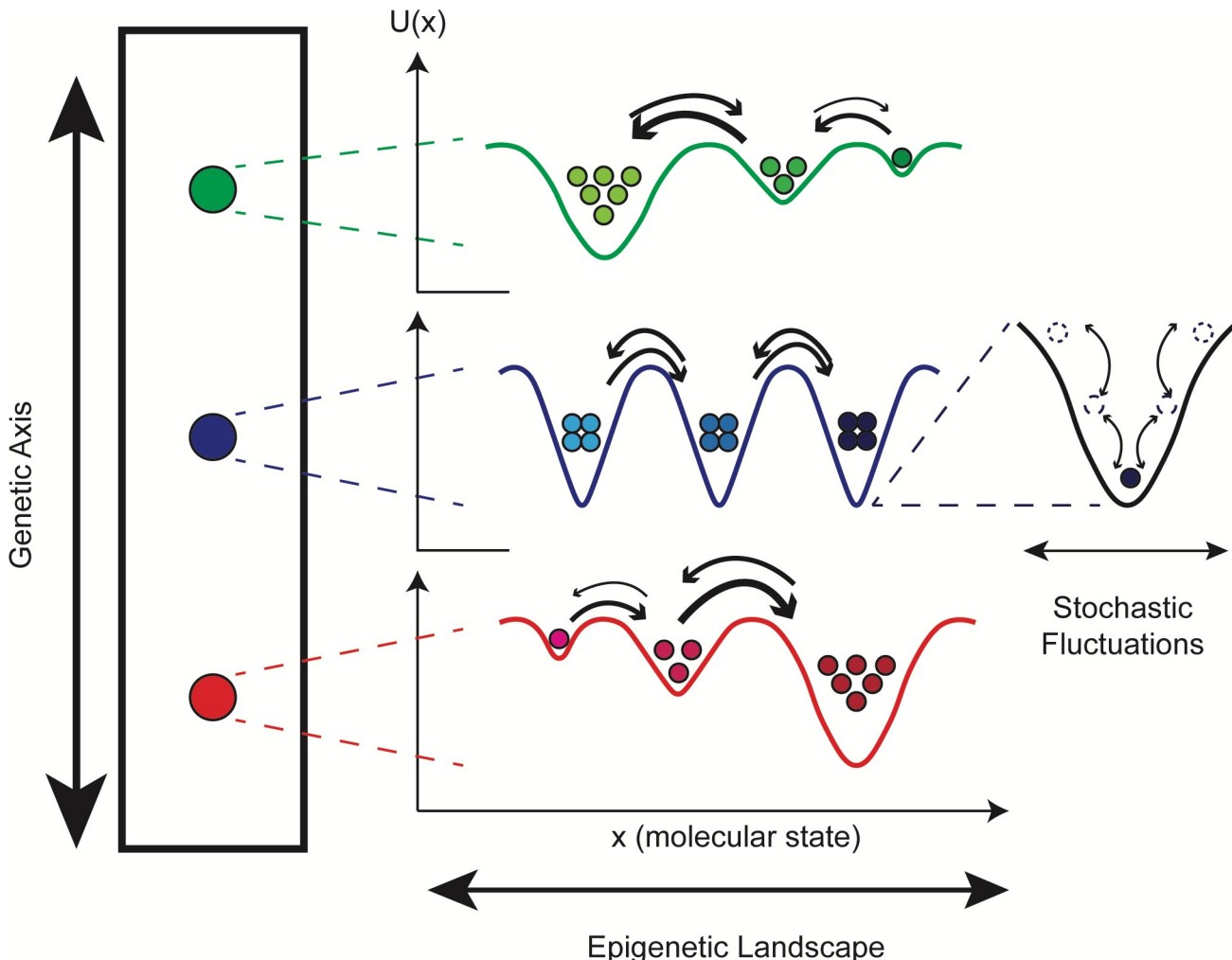

**Fig 1. Multiple levels of heterogeneity are believed to operate within tumors.** (left) The genetic "axis" defines accumulating mutational differences that have an effect on phenotype, e.g., drug sensitivity. Although depicted linearly for simplicity, note that mutational accumulation is often nonlinear, occurring instead in a branching manner. (middle) Each genetic clone has an associated epigenetic landscape, where cells are distributed across basins, known as "attractors." The topography of the epigenetic landscape is defined by the dynamical biochemical network that controls cell fate and function. Quasi-potential energy, $U(x)$, lies along the y-axis and quantifies the relative stabilities of basins; molecular state lies along the x-axis and refers to position within the high-dimensional molecular state space on which the quasi-potential energy is defined (see S1 Text for further discussion). Note that positions of basins across genetic states will not necessarily align (since mutations alter the epigenetic landscape) and they are purposely not aligned in the illustration. (right) Cell states fluctuate within epigenetic basins due to intrinsic (e.g., gene expression) and extrinsic sources of noise. Most fluctuations are minor and do not significantly change the cell state but occasionally a large fluctuation results in a barrier crossing, i.e., a phenotypic state change.

individual lineages within the family and the conditions under which each member was derived, we expect this collection of cell lines and sublines to effectively mimic the composition of a genetically and epigenetically heterogeneous tumor. We perform extensive drug response profiling of each family member, followed by genomic and transcriptomic characterization and mathematical population dynamics modeling. Using bulk genomic and single-cell RNA sequencing (scRNA-seq), we verify that the cell line versions are genetically distinct and establish quantitative benchmarks for this distinctiveness at the genomic and transcriptomic levels. Comparing genomic and transcriptomic differences across the sublines against these benchmarks, we argue that all but one of the sublines are genetically indistinct but differ epigenetically, i.e., they correspond to basins within a common epigenetic landscape. This conclusion is

further supported by stochastic simulations, which show that in all but one case, variability seen among isolated colonies of a subline in a clonal drug response assay can be explained by intrinsic randomness in cell division and death. We also detail one case where our analyses suggest that a subline harbors a distinct genetic state that appears to have emerged in the absence of selective drug pressure.

## Results

### Cell line versions and single cell-derived sublines exhibit drug response variability at the cell population level

We chose commonly used NSCLC cell line PC9 [51] as a model system for tumor heterogeneity. The PC9 cell line is characterized by an EGFR-ex19del mutation (S1A Fig), making it sensitive to inhibition of the mutant EGFR protein. We utilize 3 versions of the cell line: PC9-VU, originating from Vanderbilt University [52]; PC9-MGH, maintained at Massachusetts General Hospital [53,54]; and PC9-BR1, derived from PC9-VU and containing a known secondary resistance mutation (EGFR-T790M) obtained through dose escalation therapy in the EGFR inhibitor (EGFRi) afatinib [52]. Although it is unclear when the PC9-VU and PC9-MGH versions (originating from a common founder line) were independently established (S1B Fig), both maintain the oncogenic mutation in the EGFR gene and display sensitivity to EGFR inhibition [54,55]. In the absence of drug, PC9-VU and PC9-BR1 have essentially identical proliferation rates, while PC9-MGH grows at a slightly lower rate (Fig 2A). However, in response to the EGFRi erlotinib, the 3 cell line versions display drastically different drug sensitivities (Fig 2A): PC9-MGH exhibits substantial cell death after an initial equilibration phase (approximately 72 h), PC9-VU settles into a near-zero rate of growth, and PC9-BR1 displays insensitivity to EGFRi (as expected). These observations are consistent with the high sensitivity of PC9-MGH to erlotinib reported in Sharma and colleagues [54] and the lower sensitivity of PC9-VU that we reported previously [55,56].

We also quantified clonal drug response variability within the cell line versions using clonal fractional proliferation (cFP) [57], an assay that tracks the growth of many single cell-derived colonies over time and quantifies drug sensitivity for each colony in terms of the drug-induced proliferation (DIP) rate [55,56], defined as the stable rate of proliferation achieved after extended drug exposure (see Materials and methods). We performed cFP on each cell line version under erlotinib treatment and observed wide ranges of drug responses across colonies (S2 Fig) and substantial differences in the response distributions across versions (Fig 2B). The distribution of DIP rates for PC9-BR1 lies almost entirely in the positive DIP rate range and is clearly distinct from the others. The PC9-VU and PC9-MGH distributions have significant overlap but the PC9-MGH distribution has a marked shoulder in the negative DIP rate region, while the PC9-VU distribution extends further into the positive range. These distributions are consistent with and explain the differential drug responses observed among the cell line versions (Fig 2A): PC9-BR1 is resistant to EGFRi because its DIP rate distribution is entirely in the positive range, PC9-VU goes into a near-zero (slightly positive) growth phase because its DIP rate distribution is centered near zero, and the large shoulder in the PC9-MGH distribution explains why it exhibits significant cell death in the period immediately following drug treatment.

In addition, several single cell-derived discrete sublines (DS1, DS3, DS4, DS6, DS7, DS8, and DS9) were isolated from PC9-VU and subjected to the same analyses as above. In the absence of drug, all sublines grow at almost equal rates in culture (Fig 2C). However, in the presence of EGFRi, the sublines exhibit a wide range of responses, from positive to negative growth (Fig 2C). When overlaid with the cFP results for PC9-VU, the subline responses

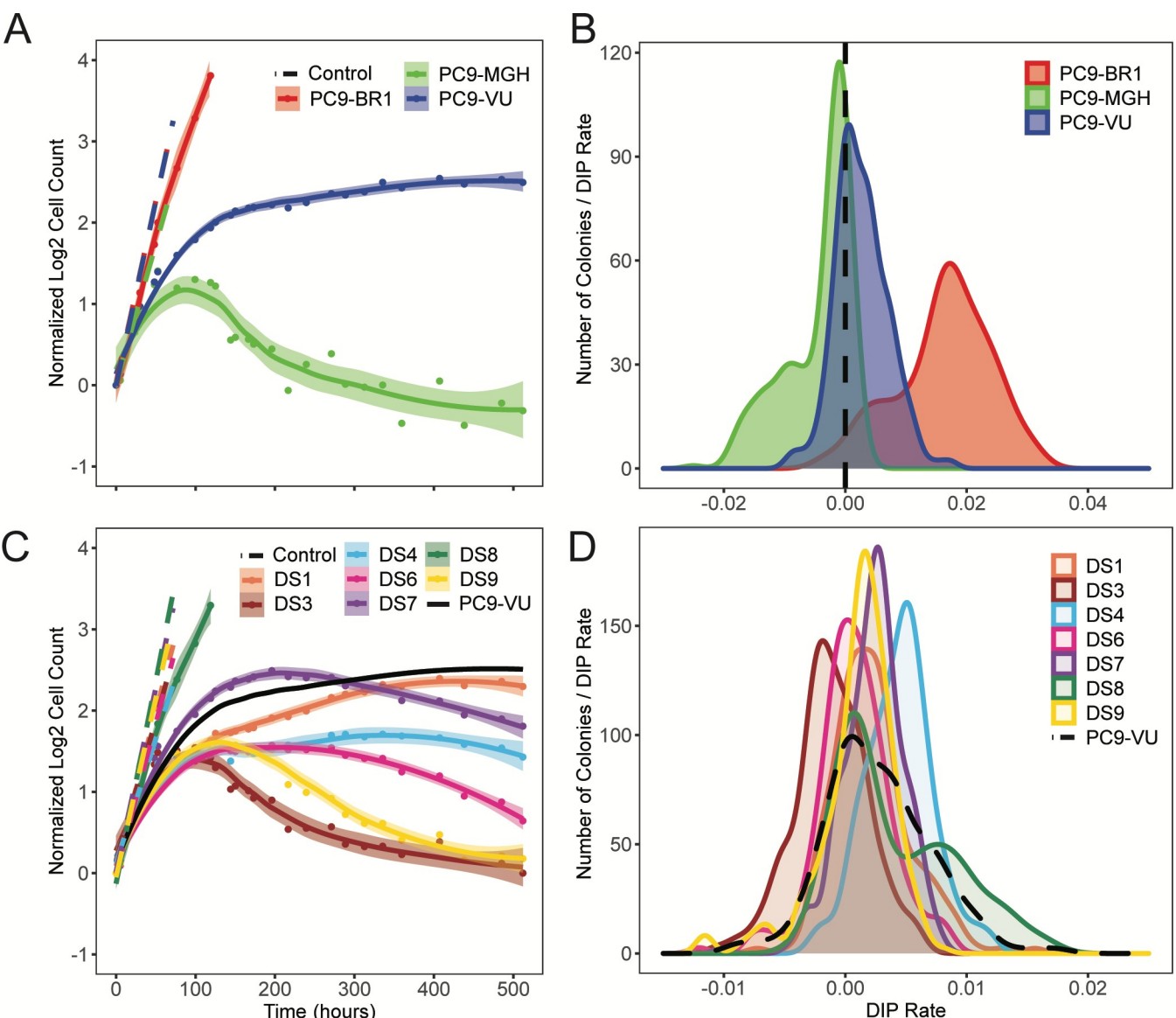

**Fig 2. Phenotypic differences among PC9 cell line versions and discrete sublines quantified in terms of drug response.** (A) Population growth curves for 3 cell line versions treated with 3 μM erlotinib for approximately 3 weeks, plus vehicle (DMSO) control. (B) DIP rate distributions compiled from single-colony growth trajectories under erlotinib treatment (3 μM) in a cFP assay. DIP rates are calculated from the growth curves 48 h postdrug addition to the end of the experiment. Dashed black lines signify zero DIP rate, for visual orientation. (C) Seven DS derived from PC9-VU were treated with 3 μM erlotinib for 3 weeks, along with vehicle control. Parental PC9-VU is included for reference. (D) DIP rate distributions from a cFP assay of the sublines in 3 μM erlotinib. Parental PC9-VU is included for reference. In A and C, dots are the means of 6 experimental replicates at each time point; solid lines are best fits to the drug response trajectories with point-wise 95% confidence intervals. In B and D, DIP rate distributions are plotted as kernel density estimates. The data underlying this figure can be found in github.com/QuLab-VU/GES_2021. cFP, clonal fractional proliferation; DIP, drug-induced proliferation; DS, discrete subline.

broadly recapitulate the observed variability seen in the parental line (S2C Fig). A notable exception is DS8, which is essentially resistant to EGFRi, having only a slightly lower proliferation rate than the fully resistant PC9-BR1 (cf. Fig 2A). We also performed cFP assays on the sublines under erlotinib treatment to quantify subclonal drug response variability (Fig 2D). Interestingly, similar to the cell line versions, we found that the sublines also exhibit distributions of DIP rates, albeit narrower than those for the cell line versions. The subline distributions have a large degree of overlap with one another, but the medians of the distributions are

statistically distinct ($p < 0.001$, Mood's median test). DS8 is again an exception, exhibiting a bimodal DIP rate distribution with a major mode centered close to zero and a large shoulder in the positive DIP rate range.

## Cell line versions differ significantly at the genetic and transcriptomic levels

Given that the PC9-VU and PC9-MGH cell line versions have been maintained separately for many years, it is virtually certain that they differ genetically due to genetic drift [48]. We also know that PC9-BR1 contains a known genetic resistance mutation and likely numerous additional mutations acquired during dose escalation. Thus, we performed bulk whole exome sequencing (WES) and scRNA-seq on the cell line versions in order to establish benchmarks for genetic variation against which we can compare the sublines.

From WES, we identify mutations (single nucleotide polymorphisms (SNPs) and insertion/deletions (InDels)) in each cell line version that pass a specified threshold for variant detection (see Materials and methods and S3A and S3B Fig) and calculate the number of mutations per chromosome (Fig 3A). We see a large amount of variability in the number of called variants between the cell line versions (average coefficient of variation (CV) per chromosome = 12.84). We also identify mutations unique to each cell line version (S3C Fig) and calculate the proportionality of unique mutations compared to the total number of mutations (Fig 3B). Although a majority of the mutations are shared among all 3 versions (approximately $10^6$ shared sequence variants), confirming that they are related through a common founder (ancestor) population, a significant number are unique: PC9-BR1 has the largest proportional representation of unique mutations, followed by PC9-MGH and then PC9-VU. Furthermore, we annotate unique mutations within each cell line version with an IMPACT score, a variant severity classifier calculated by the Ensembl Variant Effect Predictor (VEP) [58]. The IMPACT score differentiates mutations based on a variety of factors that predict whether a mutation is likely to have a phenotypic effect (see Materials and methods). Categorizing mutations into "low," "moderate," and "high" IMPACT score reveals that PC9-BR1 has many more potentially impactful mutations than PC9-MGH and PC9-VU, which have similar numbers to each other (Fig 3C). However, as a percentage, only 1% of PC9-MGH unique mutations are predicted to be impactful, compared to 11% in PC9-VU, suggesting that PC9-MGH harbors a large number of passenger mutations.

We also perform a mutational significance analysis on the unique mutations. Based on nonsynonymous-to-synonymous mutational load, genes are selected to create a mutational signature of genetic differences within each cell line version (see Materials and methods) and displayed as a heatmap. This signature does not reflect all mutated genes in the cell line versions but rather those of predicted importance, similar to the VEP IMPACT score analysis. We see that many mutations in the signature distinguish the cell line versions (Fig 3D). Additionally, we generate a literature-curated, cancer-associated gene signature that includes mutations predicted to be implicated in cancer [2,59] (see Materials and methods). Only PC9-BR1, which has a known resistance mutation (EGFR-T790M, noted as a missense mutation in the heatmap), harbors significant mutational load in this gene signature (S4A Fig). Breakdowns of mutations by type provide additional evidence that PC9-VU mutation representation is distinct and proportionally impactful (S4B Fig).

In addition to traditional variant analyses, we perform copy number variant (CNV) detection at the single-cell level. By exploring the gene expression intensity across positions of the genome, CNVs are detected as gains (red) or deletions (blue) in large chromosomal regions, making it clear which regions of the genome have different relative abundances. Since a

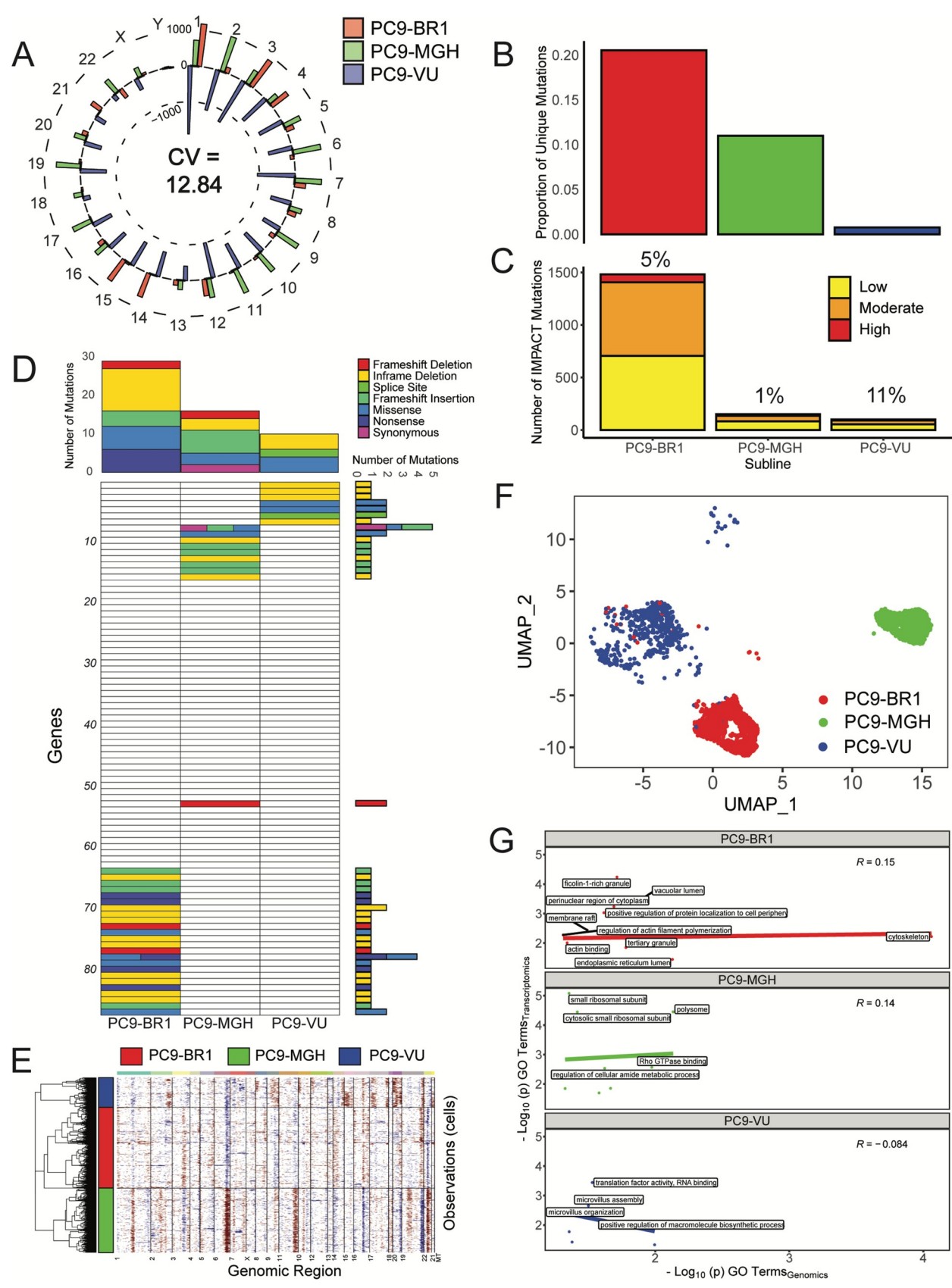

**Fig 3. Genomic and transcriptomic characterizations of PC9 cell line versions.** (A) Mean-centered mutation count by chromosome for all cell line versions. For each chromosome, versions with fewer mutations than the mean have a bar pointing inwards, while those with more mutations than the mean point outwards. Chromosome numbers are noted on the outside edge of the circle. Average CV across all chromosomes is noted. (B) Proportions of unique mutations for all cell line versions. (C) Numbers of IMPACT mutations unique to each cell line version, stratified by IMPACT classification ("low," "moderate," and "high"). Percentage of unique IMPACT mutations relative to the total number of unique mutations for each cell line version is denoted above each bar. (D) Quantification of mutational differences between cell line versions based on a signature of genes with a high nonsynonymous mutational load. Rows are ordered the same as in Fig 4D and numbered in increments of 10 for ease of reference (see Table B in S1 Text). Heatmap elements are colored based on type of mutation. Total numbers of mutations (stratified by mutation type) across genes and cell line versions are shown as bar plots to the right and above the heatmap, respectively. (E) Copy number variant detection for cell line versions. Red corresponds to amplifications, blue to deletions. The average signal across all cells in the cell line versions was used to define the baseline reference. (F) UMAP visualization of single-cell transcriptomes for cell line versions. For comparison purposes, the UMAP space is defined over all 8 PC9 samples (including cell line versions and PC9-VU sublines; see Materials and methods). (G) GO comparison analysis of unique IMPACT mutations and DEGs for cell line versions. A correlation coefficient (Pearson) was calculated for each sample. Terms with -log$_{10}$(p) > 2 on either axis are displayed on the plots. The data underlying this figure can be found in github.com/QuLab-VU/GES_2021. CV, coefficient of variation; DEG, differentially expressed gene; GO, Gene Ontology; UMAP, Uniform Manifold Approximation and Projection.

natural reference (i.e., a common ancestor) does not exist for the PC9 cell line versions, we establish the baseline as an average signal across cells in all 3 versions (as suggested by the tool instructions; see Materials and methods). Within these analytical limits, we identity several key regions that differentiate the cell line versions (Fig 3E): PC9-MGH has clear duplications at chromosomes 6, 11, and 22 and deletions at chromosomes 16 and 19, while PC9-BR1 has duplications in chromosomes 3 and 14 and deletions in chromosomes 9 and 22. This result fits with our expectation, as both PC9-MGH and PC9-BR1 have had extensive opportunity to acquire large-scale chromosomal changes while evolving in separate environments, and PC9-VU appears to be a closer genetic descendent of the PC9 founder cell line. Together, these genomic data and those presented above (Fig 3A–3D and S3 and S4 Figs) starkly illustrate the genetic differences among the cell line versions.

At the transcriptomic level, we used scRNA-seq to identify gene expression differences among the cell line versions (see Materials and methods and S5A Fig). After feature selection (S5B Fig), we use Uniform Manifold Approximation and Projection (UMAP) [60,61] to project the transcriptional states for each cell into two-dimensional space (Fig 3F and S6A Fig). We see a clear separation of cell line versions in the UMAP space, with minimal overlap (S6B Fig). Qualitatively, distances between the single-cell clusters suggest that PC9-VU and PC9-BR1 are more similar to each other than either is to PC9-MGH, which is unsurprising given that PC9-BR1 was derived from PC9-VU. These results are also supported by alternate dimensionality reduction methods (S6E and S6F Fig) and bulk RNA expression data (S7 Fig). Furthermore, to explore potential biological interpretability of these differences across cell line versions, we score each cell based on 50 hallmark gene signatures of well-defined biological states (S8 Fig). Many of these processes distinguish cell line versions (Table 1), such as IL2/STAT5 and KRAS signaling being overexpressed in PC9-MGH, while PI3K/AKT/mTOR signaling has elevated expression in PC9-VU and PC9-BR1. Interestingly, PC9-BR1 has less expression in Hedgehog signaling than PC9-MGH and PC9-VU, making it an interesting potential side effect of the increased mutational burden.

To quantify how predictive variations in the genomic states are of differential gene expression at the transcriptomic level, we compare Gene Ontology (GO) [62,63] terms associated with high consequence genetic sequence variants ("low," "moderate," and "high" IMPACT scores) and GO terms associated with significantly differentially expressed genes (DEGs; adjusted $p < 0.05$). We visualize these terms based on relative statistical significance (−log(p) for significant GO terms) and quantify the correlation (Spearman) between the genomics- and transcriptomics-derived terms for each cell line version (Fig 3G). Both PC9-BR1 and PC9-MGH have a positive correlation between terms, indicating that terms shared between data modalities tend to agree with each other. Obvious exceptions exist that are more

**Table 1. Simplified representation of hallmark gene signature scores across different members of the PC9 cell line family.** Single-cell transcriptomics data from all 8 PC9 cell line family members were subjected to a VISION functional interpretation analysis to calculate scores for single cells from 50 MSigDB hallmark gene signatures. Distributions of scores were calculated for each cell line family member-gene signature pair. Signature score distributions that show larger (↑) or smaller (↓) values compared to other cell line family members are noted. Cell line family members with the largest transcriptomic differences (PC9-MGH, PC9-BR1, DS8, and PC9-VU) were chosen for comparisons. "PC9-VU (no DS8)" includes the parental PC9-VU and DS3, DS6, DS7, and DS9 subline distributions, since they all have a large degree of transcriptomic overlap.

| PC9-MGH | PC9-BR1 | DS8 | PC9-VU (no DS8) |
|---|---|---|---|
| ↑ Angiogenesis | ↑ Cholesterol Homeostasis | ↑ Allograft Rejection | ↑ Hedgehog Signaling |
| ↑ Apical Surface | ↓ Hedgehog Signaling | ↑ Androgen Response | ↓ IL2/STAT5 Signaling |
| ↑ Bile Acid Metabolism | ↓ Xenobiotic Metabolism | ↑ Complement | ↓ WNT/β-catenin Signaling |
| ↑ IL2/STAT5 Signaling | | ↑ DNA Repair | |
| ↑ KRAS Signaling | | ↑ Interferon α/γ Response | |
| ↑ WNT/β-catenin Signaling | | ↑ Unfolded Protein Response | |
| ↓ Allograft Rejection | | ↓ P53 Pathway | |
| ↓ Coagulation | | | |
| ↓ Interferon α/γ Response | | | |
| ↓ Pancreas β Cells | | | |
| ↓ PI3K/AKT/MTOR Signaling | | | |

statistically significant for one data modality (top-left and bottom-right corners of the plots), but both the existence of terms for both modalities and the moderate correlation indicate the connection. Notably, PC9-VU has a slightly negative correlation. We also use a semantic similarity metric [64] to compare the 2 sets of GO terms. For each version, we calculate pairwise similarity scores between GO terms for genetic variants and DEGs and obtain an aggregate score between 0 and 1, with 1 indicating that DEGs at the transcriptomic level can be perfectly explained by variations at the genomic level and vice versa (see Materials and methods and S1 Text for details). Relative to a randomized baseline, we see elevated semantic similarity scores for PC9-BR1 in the "Biological Process" (BP) GO category and for PC9-VU in the "Molecular Function" (MF) category (S9 Fig). We also see significant semantic similarity scores for all 3 versions relative to baseline in the "Cellular Component" (CC) category. Taken together, these results (Fig 3G and S9 Fig) indicate a strong connection between mutations in the genomes of the cell line versions and expression at the transcriptomic level. We can use these results as benchmarks for ascertaining whether transcriptomic differences seen among other samples (e.g., single cell-derived sublines; see next subsection) are rooted in genetic differences, like in the cell line versions, or are likely nongenetic in origin.

## One PC9-VU subline is genetically distinct, while all others are transcriptomically distinct from each other

We performed the same genomic and transcriptomic experiments as above on 5 PC9-VU sublines (DS3, DS6, DS7, DS8, and DS9) that exhibit differential responses to EGFRi as evidenced by their DIP rate distributions (see Fig 2D): DS3 has a peak in the negative DIP rate range, DS6 has a peak close to zero, DS7 and DS9 have peaks in the positive DIP rate range and nearly overlapping distributions, and DS8 stands out as an obvious outlier with a bimodal DIP rate distribution. Analysis of the total numbers of mutations by chromosome (Fig 4A) shows significantly less variability in the variant count for the sublines relative to the cell line versions (average CV per chromosome = 6.27 versus 12.84; cf. Fig 3A). Additionally, unlike the cell line versions, most sublines exhibit similar proportions of unique mutations (Fig 4B; S3D Fig for full overlap) and numbers of IMPACT mutations (Fig 4C). The clear exception is DS8, which has more unique mutations and more than twice the number of predicted impactful mutations

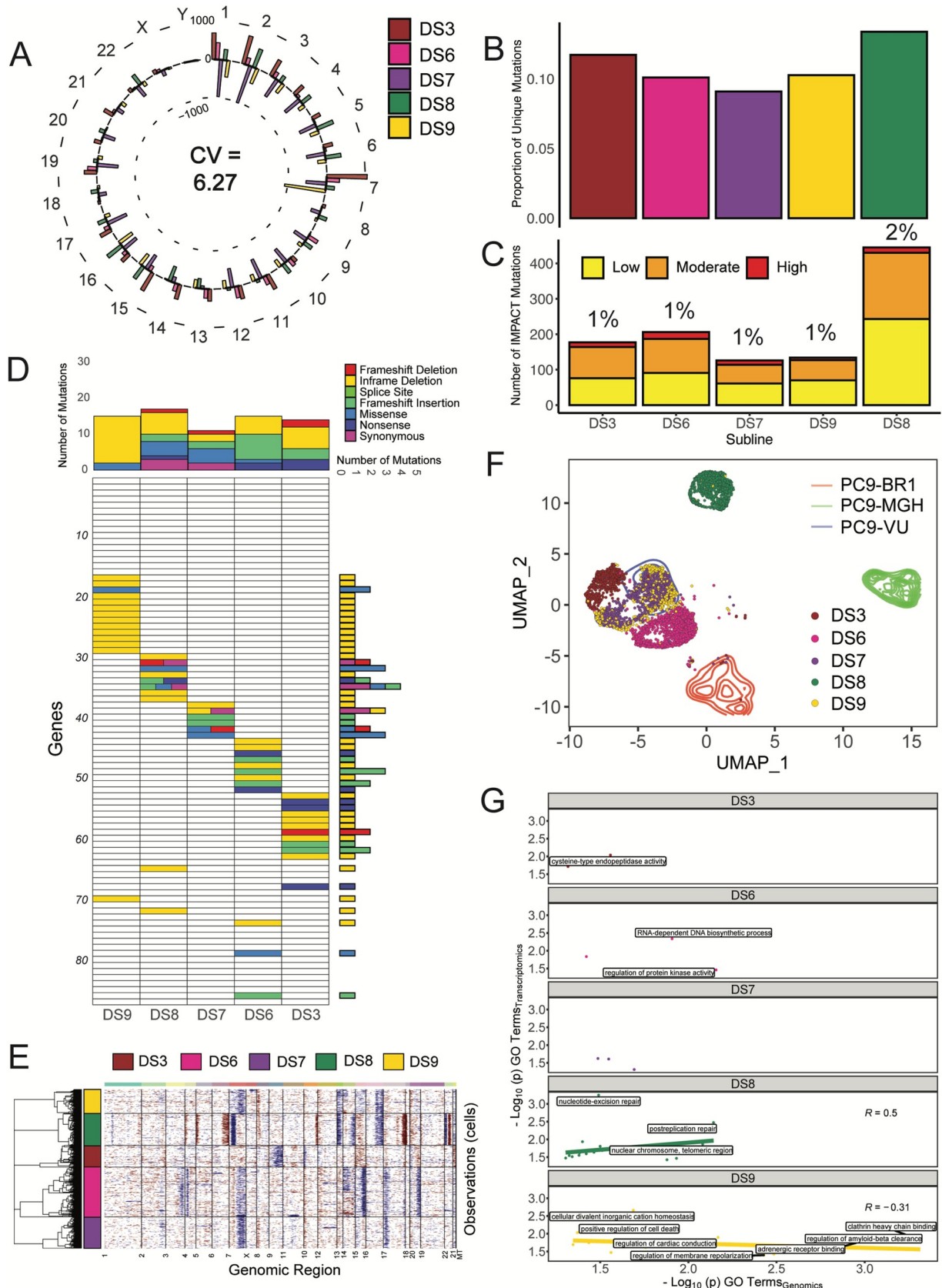

**Fig 4. Genomic and transcriptomic characterizations of PC9-VU discrete sublines.** (A) Mean-centered mutation count by chromosome for 5 (of the 7) sublines. Average CV across all chromosomes is noted. (B) Proportions of unique mutations in each subline. (C) Numbers of IMPACT mutations unique to each subline, stratified by IMPACT classification ("low," "moderate," and "high"). Percentage of unique IMPACT mutations relative to the total number of unique mutations in each subline is denoted above each bar. (D) Quantification of mutational differences between sublines based on the same gene signature as for the cell line versions (Fig 3D). Rows are ordered the same as in Fig 3D and numbered in increments of 10 for ease of reference (see Table B in S1 Text). Heatmap elements are colored based on type of mutation. (E) CNV detection for sublines. Red corresponds to amplifications, blue to deletions. PC9-VU was used as the baseline reference to compare sublines. (F) UMAP visualization of single-cell transcriptomes for sublines plotted in the common UMAP space of all 8 PC9 samples (including cell line versions and PC9-VU sublines; see Materials and methods). (G) GO comparison analysis of unique IMPACT mutations and DEGs for sublines. A correlation coefficient (Pearson) was calculated for each sample. Terms with -log$_{10}$(p) > 2 on either axis are displayed on the plots. Samples without a correlation line or coefficient (DS3, DS6, and DS7) did not meet the minimum threshold of data points to compute a correlation. The data underlying this figure can be found in github.com/QuLab-VU/GES_2021. CNV, copy number variant; CV, coefficient of variation; DEG, differentially expressed gene; GO, Gene Ontology; UMAP, Uniform Manifold Approximation and Projection.

compared to the other sublines. Mutational significance analysis (using the same genomic signature as for the cell line versions; see Fig 3D) shows similar numbers of total and impactful mutations in the sublines (Fig 4D), while DS8 has the largest and most diverse set of mutation types. This is true for the cancer-associated genes as well (S4A Fig). The sublines also exhibit a nearly identical mutation class distribution (S4B Fig). Interestingly, CNVs in the sublines show a slightly more nuanced result (Fig 4E; background PC9-VU in S10 Fig): DS8 has major amplifications (chromosomes 6, 17, and 22) and deletions (chromosomes 7, 14, and 22) and some minor alterations not shared with other sublines; DS3 is missing a deletion present in chromosome 7 in DS6, DS7, and DS9 and also has a unique deletion in chromosome 9; and minor additional sharing is seen among other sublines, such as DS8 and DS9 (deletions in chromosomes 13 and 16). On the whole, DS8 has the clearest cases of unique CNVs in the sublines, while the other sublines remain largely similar except for a few instances. Taken together, with the exception of DS8, these genomic data (Fig 4A–4E and S3 and S4 Figs) illustrate that there is significantly less genomic variability among the PC9-VU sublines than among the cell line versions. It is also important to note that DS8 does not harbor the same resistance conferring mutation (EGFR-T790M) that PC9-BR1 does (S4 Fig), indicating a different (unknown) resistance mechanism is at play (see S1 Text for further discussion).

Comparing single-cell transcriptomes (Fig 4F), we see distinctions among the sublines but to a much lesser extent than among the cell line versions, except for DS8, which, as in the genomics data, is a clear exception (S7C and S7D Fig). Qualitative distances between single-cell clusters show virtually no separation between DS7 and DS9, small but clear separations between DS3, DS6, and DS7/DS9, and a large separation between DS8 and the other sublines. We also see that DS3, DS6, DS7, and DS9 substantially overlap with the PC9-VU region of the UMAP space (Fig 4F). These observations are largely consistent with the clonal drug responses observed in the cFP assays (i.e., the DS7 and DS9 distributions are almost identical; the DS3 and DS6 distributions are distinct from each other and from DS7/DS9; the DS8 distribution stands apart from the others in being bimodal with a large shoulder extending beyond the upper range of the PC9-VU parental distribution; and the DS3, DS6, DS7, and DS9 distributions overlap substantially with the PC9-VU distribution; see Fig 2D). However, despite the clear separations of the clusters, we do see slight overlaps at the boundaries between the transcriptomic features for DS3 and DS7/DS9 and between DS6 and DS7/DS9, suggesting the possibility of phenotypic transitions occurring between these states. We also see a very small number of DS9 cells (<2%) that overlap with DS8. This is an interesting observation, which could have multiple possible explanations (see S1 Text and S11 Fig for further discussion). In terms of biological interpretability, DS8 has a larger hallmark gene signature score than the other sublines (and cell line versions) for DNA repair, unfolded protein response, and androgen response, while it has a slightly lower score in the p53 pathway (Table 1 and S8 Fig). There are no clear cases where other sublines had a significantly larger hallmark gene signature score.

Statistical comparisons between GO terms associated with high consequence genetic variants and DEGs support a connection between genomics and transcriptomics in DS8 but not in the other sublines (Fig 4G). Although DS3, DS6, and DS7 have a few terms significantly enriched and shared between data modalities, there were not enough data points to compute a correlation. DS9 is an interesting case, where many terms were similar between the modalities but showed a negative correlation (more so than PC9-VU; cf. Fig 3G). In terms of semantic similarity, we see mixed results across sublines (S9 Fig): DS8 has high scores for both BP and MF GO categories but not for CC; DS3, DS6, and DS7 have low scores for BP and MF; DS6 also has a low score for CC but DS3 and DS7 have high scores; and DS9 has a high score for the BP category but low scores for the others. Note that based on the number of GO terms in each category (BP: 12,272, MF: 4,165, and CC: 1,740), we consider BP to be the most predictive of the three, followed by MF and then CC (see Materials and methods). Taken together, these results suggest a strong connection between genomics and transcriptomics in DS8 but weaker or nonexistent connections in the other sublines.

## Stochastic birth-death simulations suggest most PC9-VU sublines are epigenetically monoclonal, while one is polyclonal

The PC9-VU sublines exhibit variability in drug response not just at the population level (Fig 2C) but also at the subclonal level, as evidenced by variable colony growth in cFP assays (Fig 5A and 5E and S12A Fig) and quantified as distributions of DIP rates (Fig 2D). To explore the origin of this subclonal variability, we performed stochastic simulations [65] on a simple birth-death model of cell proliferation to ascertain whether intrinsic noise in division/death decisions alone is sufficient to explain experimental observations (see Materials and methods). We performed a battery of in silico cFP assays, where untreated single cells grow into colonies of variable size at a fixed proliferation rate (division rate constant–death rate constant) and are then treated with drug, modeled by reducing the proliferation rate. Colony sizes are tracked over time (Fig 5B and S12B Fig) and DIP rate distributions are calculated and statistically compared against experimental distributions (Fig 5C and S12C Fig). We repeated this procedure for a wide range of division and death rate constants to identify ranges of parameter values that can statistically reproduce experimental DIP rate distributions ($p > 0.05$, bootstrapped Anderson–Darling (AD) test). For all sublines (except DS8, see next paragraph), we find ranges of parameter values that are physiologically plausible (Fig 5D and S12D Fig). This result is consistent with the view that these sublines (DS1, DS3, DS4, DS6, DS7, and DS9) are monoclonal, i.e., experimental DIP rate distributions can be reproduced with a birth-death model comprising a single cell state (one division and one death rate constant) simulated stochastically.

In contrast, DS8 is an exception once again, displaying greater variability than the other sublines in cFP colony growth rates (Fig 5E) and a bimodal DIP rate distribution (Fig 2D). We performed stochastic simulations on an expanded version of the birth-death model comprising 2 cell states with distinct division and death rate constants (see Materials and methods and S1 Text). As with the other sublines, we performed in silico cFP assays (Fig 5F) and compared the simulated DIP rate distributions against the bimodal distribution seen experimentally (Fig 5G). We again find physiologically feasible ranges of rate parameter values that can statistically reproduce the experimental result (Fig 5H). Thus, these results provide strong evidence that DS8 harbors at least 2 distinct cell states. It is important to note that the mathematical model is agnostic as to whether these 2 states are genetically or epigenetically distinct; it simply states that they have distinct division and death rate constants.

## Discussion

Many modern cancer therapies focus on targeting specific genetic mutations within a tumor. Recent studies have shown that a complex interplay between genetic and nongenetic factors

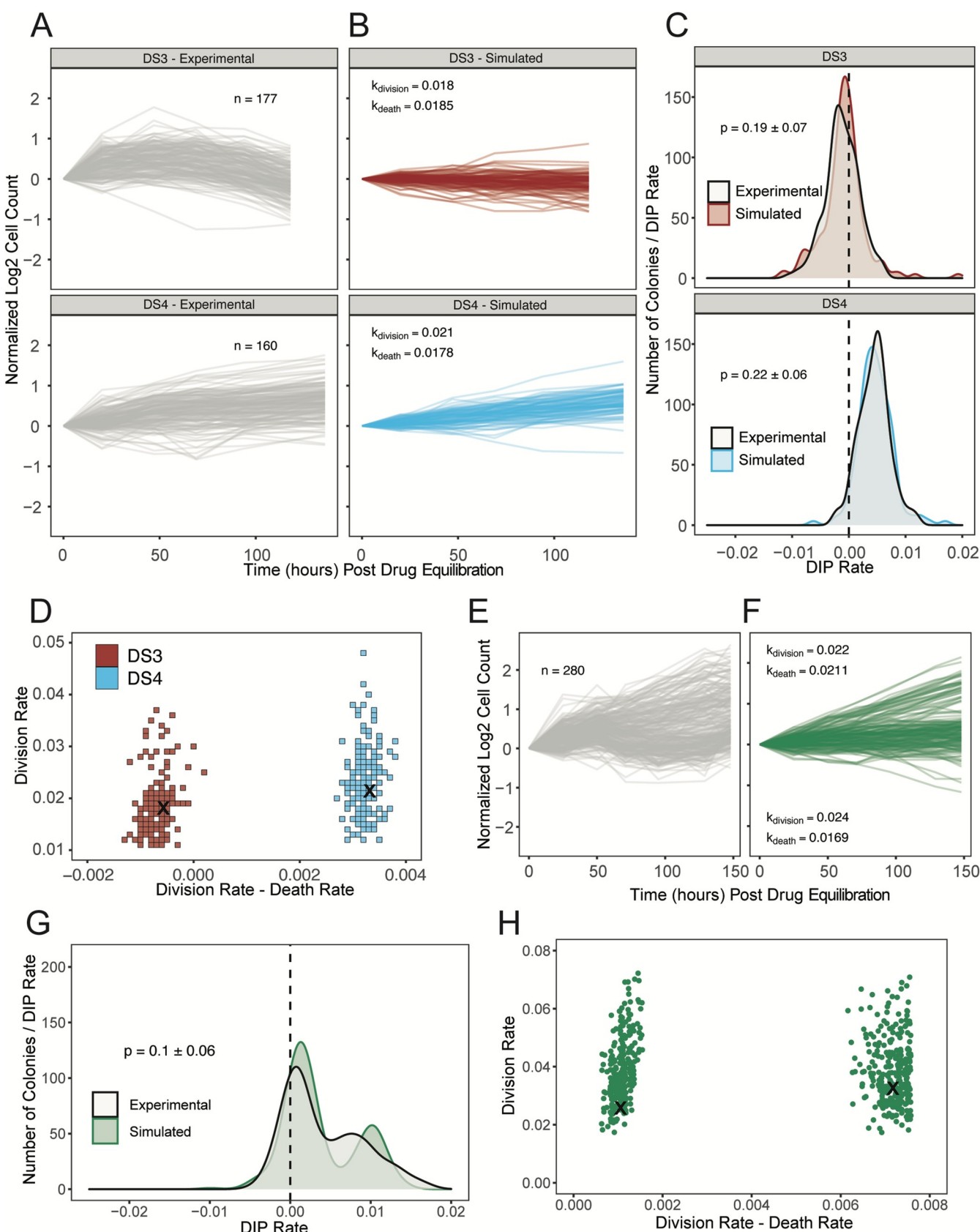

**Fig 5. Stochastic simulations of a simple birth-death model reproduce DIP rate distributions from PC9-VU sublines.** (A) Experimental cFP time courses for 2 representative sublines (DS3 and DS4) in response to 3 μM erlotinib (same data used to generate the corresponding DIP rate distributions in Fig 2D). Each trace corresponds to a single colony, normalized to 72 h postdrug treatment. Only colonies with initial cell counts greater than 50 at the time of treatment are shown. (B) In silico cFP time courses from a one-state model with division and death rate constants that closely reproduce the experimental time courses in A. Trajectories are normalized to the time at which the simulated drug treatment was initiated, simulated cell counts are plotted only at experimental time points, and only colonies with initial cell counts greater than 50 at the time of simulated drug treatment are shown. (C) Comparison of experimental and simulated DIP rate distributions from time courses in A and B. Distributions are compared statistically using the AD test [66]. Dashed black line signifies zero DIP rate, for visual orientation. (D) Parameter scan of division and death rate constants for the 2 sublines in A–C. For each pair of rate constants, the same number of model simulations were run as the associated experimental cFP time courses in A. DIP rates were calculated and compiled into a distribution and then statistically compared against the corresponding experimental DIP rate distribution using the AD test. All parameter pairs with $p < 0.05$ (see Materials and methods) are colored white, indicating lack of statistical correspondence to experiment. "×" denotes a division and death rate constant pair used in B. (E) Same as A but for subline DS8. (F) Same as B but for DS8 using a two-state model. (G) Same as C but for DS8. (H) Same as D but for DS8 using a four-dimensional (2 division-death rate constant pairs) parameter scan and projected into 2 dimensions. "×" denotes parameter values used to generate the simulated DIP rate distribution in F. The data underlying this figure can be found in github.com/QuLab-VU/GES_2021. AD, Anderson–Darling; cFP, clonal fractional proliferation; DIP, drug-induced proliferation.

likely plays a key role in the failure of targeted treatments [8,53]. Here, we investigated genetic and nongenetic sources of variability in an in vitro tumor heterogeneity model comprising multiple versions (VU, MGH, and BR1) and single cell-derived sublines of the NSCLC cell line PC9 that exhibit a wide range of different responses to EGFR inhibition (Fig 2). Given their histories and how each was derived, we had good reason to believe that the cell line versions were genetically distinct. This was validated using WES and CNV detection, which showed significant mutational differences among them (Fig 3A–3E). Distinct transcriptomic features were also identified by scRNA-seq (Fig 3F), and connections to the underlying genetic states were established by a comparison between GO terms enriched in each data modality (Fig 3G and S9 Fig). We then isolated 7 sublines from PC9-VU that exhibited differential responses to EGFR inhibition (Fig 2C). Clonal drug response assays (Fig 2D) and scRNA-seq analysis (Fig 4F) showed significant overlap with the PC9-VU parental cell line. WES and CNV detection revealed substantially less genomic variability among 6 of the 7 sublines relative to the cell line versions (Fig 4A–4E) and GO similarity analysis indicated a weak connection, if any, between genomic and transcriptomic states in these sublines (Fig 4G and S9 Fig). For the other subline, DS8, the results were dramatically different: DS8 harbors significantly more unique and IMPACT mutations than the other sublines (Fig 4B–4D), there is clearer evidence for copy number variation (Fig 4E), its single-cell transcriptomic state is substantially distinct from the other sublines (Fig 4F), and it displays a much stronger connection between genomic and transcriptomic states (Fig 4G and S9 Fig). Finally, stochastic simulations revealed that colony growth dynamics for 6 of the 7 sublines can be explained as a population with a single cell state experiencing probabilistic division/death decisions (Fig 5A–5D). For DS8, a second cell state had to be included in the model to reproduce the bimodal DIP rate distribution observed experimentally (Fig 5E–5H).

In order to interpret our results, we utilize the theoretical framework for tumor heterogeneity discussed previously [19,20,37–39] (Fig 1). As explained, in this view of tumor heterogeneity, tumors may comprise multiple genetic states, each of which has an associated epigenetic landscape with ≥1 quasi-potential energy basins corresponding to phenotypic states, across which cells can transition driven by intrinsic (e.g., gene expression) or extrinsic noise sources. Within our in vitro tumor model, the PC9 cell line versions (VU, MGH, and BR1) correspond to the different genetic states. We assert that 4 of the PC9-VU sublines (DS3, DS6, DS7, and DS9), based on their genomic similarity (Fig 4A–4E), transcriptomic distinctiveness (Fig 4F), weak genetic-to-transcriptomic correspondence (Fig 4G), and monoclonality (Fig 5A–5C), likely correspond to basins within the epigenetic landscape associated with the PC9-VU genetic state. In contrast, DS8 appears to harbor a distinct genetic state that emerged out of PC9-VU at some point in the past in the absence of selective drug pressure. We come to this

conclusion based on its resistance to EGFRi (Fig 2C), genomic (Fig 4A–4E) and transcriptomic (Fig 4F) distinctiveness from the other sublines and all 3 cell line versions, strong genomic-to-transcriptomic correspondence (Fig 4G), apparent polyclonality (Fig 5E–5H), and lack of the resistance-conferring mutation (EGFR-T790M) found in PC9-BR1 (S4A Fig). Note that it remains an open question as to whether this new genetic state emerged prior or subsequent to establishment of the DS8 subline (see S1 Text and S11 Fig for further discussion) and what the mechanistic basis for its drug resistance is. These are important questions and areas of future investigation. We summarize our conclusions in a schematic illustrating the different sources of cell state variability we hypothesize are operating within the PC9 family of cell lines and sublines (Fig 6).

This view of tumor heterogeneity, as a three-tiered amalgamation of genetic, epigenetic, and stochastic factors, is not yet broadly accepted within the cancer research community [39], largely because of a lack of strong experimental evidence to support it. A primary goal of this work has been to provide such evidence. However, we believe that numerous reports in the literature are also consistent with this view. For example, Ben-David and colleagues [48] showed that numerous "strains" (comparable to our cell line versions) of human cancer cell lines, obtained from different institutions, display extensive genetic heterogeneity. Moreover, genetically similar strains exhibited similar transcriptomic signatures and drug response profiles. Thus, they argued that cancer cell lines can drift genetically when kept in culture independently, consistent with our results for the PC9 cell line versions (Fig 3). Our conclusion that the drug-resistant DS8 genetic state emerged spontaneously from PC9-VU in the absence of drug (Fig 6) aligns with observations by Ramirez and colleagues [2] and Hata and colleagues [31], who independently reported diverse resistance-conferring mutations arising in both untreated and drug-treated PC9-MGH clones. Shaffer and colleagues [67] described a transient, transcriptionally encoded preresistance state in 2 BRAF mutant melanoma cell lines that cells can transition into and out of in the absence of drug. We hypothesize that this preresistant state may constitute a basin within a BRAF mutant melanoma epigenetic landscape, similar to our single cell-derived sublines (DS3, DS6, DS7, and DS9) that we contend occupy the PC9-VU epigenetic landscape. The veracity of this hypothesis depends on how long cells reside in the preresistant state (its stability) and, correspondingly, whether the state is heritable by progeny cells. These are interesting questions, about which there remains much debate [68,69], and potential avenues of future investigation.

It is now abundantly clear that a focus on tumor genetics alone cannot solve the complex problems of tumor progression, metastasis, and treatment failure that continue to plague clinical oncology [70,71]. The view of tumor heterogeneity advocated in this work offers an alternative to the traditional gene-centric view and may transform how we understand and treat the disease. For example, that each genetic state has an associated epigenetic landscape with potentially numerous accessible phenotypic states may explain why targeted drug treatments eventually fail in almost all cases [72]: Some of these phenotypes may have molecular compositions that enable their survival under drug treatment. Cells preexisting in these states (e.g., the preresistance state of Shaffer and colleagues [67]), and those that escape into them upon drug addition, may act as a refuge from which genetic resistance mutations can arise [33,54]. Alternative treatments based on targeting cancer stem cells (CSCs) [73] have also been proposed but have so far proven unsuccessful [74]. This could be because CSCs correspond to shallow basins within an epigenetic landscape; killing cells in this basin does not eradicate the basin, hence leaving it available to be repopulated by cells from "adjacent" basins. It has been suggested that a better approach, termed "targeted landscaping" [33,50], is to use drugs in combination or in sequence to alter the topography of a landscape to favor drug-sensitive states over drug-tolerant states [75]. The feasibility of such an approach is supported by multiple studies showing

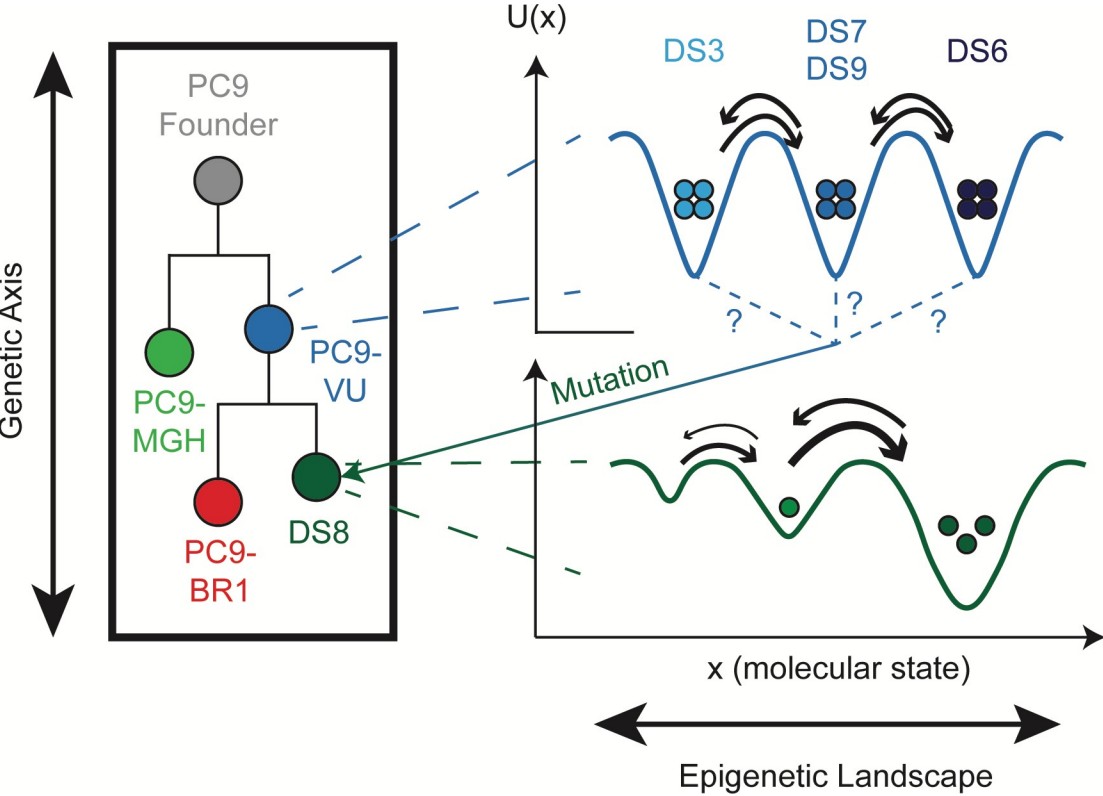

**Fig 6. Schematic interpretation of the results of our analyses on the PC9 cell line family.** Cell line versions PC9-MGH (light green) and PC9-VU (blue) represent different genetic clones within our in vitro tumor model that emerged from a common PC9 founder clone (gray). Each genetic clone has an associated epigenetic landscape, including PC9-VU, which has at least 3 distinct basins corresponding to the sublines DS3, DS7/DS9, and DS6. Cell line version PC9-BR1 (red) was derived from PC9-VU via drug-induced genetic clonal selection in EGFRi, while DS8 (dark green) independently acquired a genetic resistance mutation in the absence of selective drug pressure. However, it remains unclear from which basin DS8 arose (signified by question marks) and whether the resistance mutation was acquired before or after the subline was established (see S1 Text and S11 Fig). Note that PC9-VU is depicted as lying closer in the phylogenetic tree to the PC9 founder line than PC9-MGH because PC9-VU appears to be a closer genetic descendent of the founder, based on our genomic data. Likewise, DS8, which is believed to have emerged only recently, is depicted closer than PC9-BR1 to the common ancestor PC9-VU. EGFRi, EGFR inhibitor.

that resistance to one drug can confer sensitivity to another, known as "collateral sensitivity" [76–78], and that sequential drug applications can often be more effective than up-front drug combination treatments [79,80]. Looking forward, one can envision future cancer treatment regimens involving genetic profiling of a tumor to identify dominant genetic states, followed by characterization of the associated epigenetic landscapes using single-cell experimentation and computational modeling [46,81–89]. Large-scale in vitro and in silico drug screens could then be performed to devise personalized treatments for patients that can be tested in vivo before being administered clinically. By leveraging state-of-the-art technologies and currently available drugs to tackle tumor heterogeneity at the genetic, epigenetic, and stochastic levels, this approach may finally give researchers a leg up in the long-standing War on Cancer.

## Materials and methods

### Cell culture and reagents

PC9-VU and PC9-BR1 were obtained as a gift from Dr. William Pao (Roche, formerly of Vanderbilt University Medical Center). PC9-MGH was obtained as a gift from Dr. Jeffrey

Settleman (Massachusetts General Hospital). PC9-VU, PC9-MGH, and PC9-BR1 were individually fluorescently labeled with histone H2B conjugated to monomeric red fluorescent protein (H2BmRFP), as previously described [33,55,56,90,91]. The PC9 cell line versions and derivatives were cultured in Roswell Park Memorial Institute (RPMI) 1640 Medium (Corning Inc., Corning, NY, USA) with 10% fetal bovine serum (Thermo Fischer Scientific, Waltham, MA, USA). Cells were incubated at 37˚C, 5% $CO_2$, and passaged twice a week using TrpLE (Thermo Fischer Scientific, Waltham, MA, USA). Cell lines and sublines were tested for mycoplasma contamination using the MycoAlert mycoplasma detection kit (Lonza, Basel, Switzerland), according to manufacturer's instructions, and confirmed to be mycoplasma free. Cell line identity was confirmed using mutational signatures in WES. Approximately 99% of the PC9-VU mutations were also seen in PC9-MGH, but PC9-MGH exhibited a large number of unique and total mutations independently (S3C Fig). Additionally, using the *SNPRelate* R package (version 1.18.1), sample genotypes were projected into low dimensional space and clustered based on similarity (S13 Fig). PC9-VU and PC9-MGH clustered together and were represented closely in reduced dimensional space, further suggesting that they arose from the same parent cell population. Erlotinib was obtained from Selleck Chemicals (Houston, Texas) and solubilized in dimethyl sulfoxide (DMSO) at a stock concentration of 10 mM and stored at −20˚C. Cell lines were originally stored at −80˚C, then moved into liquid nitrogen.

## Derivation of single-cell sublines

The PC9-VU sublines were generated by limiting dilution of the parental cell line in 96-well plates. Wells with single cells were expanded for multiple weeks until large enough to be saved as frozen cell stocks. A single stock of each subline was brought back into culture, passaged for 2 weeks, and used for drug response experiments. After 3 to 4 weeks of continued passaging, sublines were used for WES, bulk RNA sequencing, and scRNA-seq experiments. Since sublines were isolated from PC9-VU, they retained the same H2BmRFP nuclear label as cell line versions. An overlap analysis of PC9-VU and derived sublines (S3E Fig) shows the fewest number of unique mutations is in the parental PC9-VU and, therefore, a strong genetic overlap between it and the sublines.

## Population-level DIP rate assay

Cells were seeded in black, clear-bottom 96-well plates (Corning Inc., Corning, NY, USA) at a density of 2,500 cells per well with 6 replicates for each sample. Plates were incubated at 37˚C and 5% $CO_2$. After cell seeding, drug was added the following morning and changed every 3 days until the end of the experiment or confluency. Untreated samples were allowed to grow in DMSO-containing media until confluency, with media changes every 3 days. Plates were imaged using automated fluorescence microscopy (Cellavista Instrument, Synentec, Elmshorn, Germany). Twenty-five nonoverlapping fluorescent images (20X objective, $5 \times 5$ montage) were taken twice daily for a total of 500 hours or until confluency. Cellavista image segmentation software (Synentec) was utilized to calculate nuclear count (i.e., cell count) per well at each time point (Source = Cy3, Dichro = Cy3, Filter = Texas Red, Emission Time = 800 μs, Gain = 20×, Quality = High, Binning = $2 \times 2$). Cell nucleus count across wells was used to calculate mean and 95% confidence intervals and normalized to time of drug treatment. Data was visualized using the *ggplot2* R package (version 3.2.0).

## Clonal fractional proliferation assay

We modified the original cFP assay, which tracks multiple colonies in a single well of a plate [57]. Instead, here we flow sorted single cells into a black, clear-bottom 384-well plate (Greiner

Bio-One, Kremsmünster, Austria) using fluorescence-activated cell sorting (FACS Aria III, RFP$^+$). Plates were incubated at 37˚C, 5% $CO_2$, and cells were allowed to grow into small colonies over 8 days in complete media (no media change). Drug was then added and changed every 3 days. Plates were imaged using the Cellavista Instrument (Synentec). Nine nonoverlapping fluorescent images (3 × 3 montage of the whole well at 10X magnification) were taken once daily for a total of 7 days. Cellavista image segmentation software (Synentec) was utilized to calculate nuclear count (i.e., cell count) per well at each time point (Source = Cy3, Dichro = Cy3, Filter = Texas Red, Emission Time = 800 μs, Gain = 20×, Quality = High, Binning = 2 × 2). Depending on the number of wells that passed quality control thresholding (at least 50 cells per colony at the time of treatment), 160 to 280 replicates were included for each sample. DIP rates were calculated from 48 h posttreatment to the end of the experiment using the *lm* function in R. DIP rates for each sample were combined and plotted as a kernel density estimate. Mood's median test [92] was performed to determine statistical difference between subline DIP rate distributions using the *RVAideMemoire* R package. Data were visualized using the *ggplot2* package.

## DNA bulk whole exome sequencing

**Data acquisition.** Genomic DNA was extracted using the DNeasy Blood and Tissue Kit (Qiagen, Hilden, Germany), according to the manufacturer's protocol. Libraries were prepared using 150 ng of genomic DNA by first shearing the samples to a target insert size of 200 bp. Illumina's TruSeq Exome kit (Illumina, Cat: 20020615) was utilized per manufacturer's instructions. The samples were then captured using the Integrated DNA Technologies (Coralville, IA, USA) xGen Exome Research Panel v1.0 (IDT, Cat: 1056115). The resulting libraries were quantified using a Qubit fluorometer (ThermoFisher Scientific, Waltham, MA, USA), Bioanalyzer 2100 (Agilent Technologies, Santa Clara, CA, USA) for library profile assessment, and qPCR (Kapa Biosystems, Wilmington, MA, USA, Cat: KK4622) to validate ligated material, according to the manufacturer's instructions. The libraries were sequenced using the NovaSeq 6000 with 150 bp paired-end reads. RTA (version 2.4.11; Illumina, San Diego, CA, USA) was used for base calling and sequence-specific quality control analysis was completed using MultiQC v1.7. Reads were aligned to the University of California, Santa Cruz (UCSC) hg38 reference genome using BWA-MEM [93] (version 0.7.17) with default parameters.

**Genomic mutational analysis.** Mutation analysis for SNPs and InDels was performed using an in-house variant calling pipeline based on the Genome Analysis Toolkit (GATK, Broad Institute) recommendations. Duplicate reads were marked and replaced using PICARD (Broad Institute). Base recalibration and variant calling were performed using GATK version 3.8 (Broad Institute). Variants were selected and filtered based on gold standard SNPs and InDels, as well as a hard filtration according to GATK recommendations (SNPs: QD<2, QUAL<30, SOR>3, FS>60, MQ<40, MQRankSum<−12.5, ReadPosRankSum<−8; InDels: QD<2, QUAL<30, FS>200, ReadPosRankSum<−20). Total variants were counted using VCFtools [94] (version 0.1.15). Sequencing metrics were calculated using vcfR [95] (version 1.8.0). These metrics included read depth, mapping quality, and a Phred quality score [96,97]. Variants were annotated by VEP [58] (Ensembl Genome Browser version 95, available at ensembl.org) with multiple indicators, including chromosome name, gene symbol, mutation class, mutation type, and IMPACT rating. Mutation class corresponds to whether a variant is a substitution (single nucleotide variant (SNV), sequence alteration) or Indel (insertion, deletion, or both—also referred to as indel). Type corresponds to the result of a variant on the amino acid sequence: synonymous (no effect), missense (codon change), nonsense (codon stop or start), splice site (boundary of intron and exon), or a shift in frame (inframe or

frameshift). IMPACT rating is a subjective classification of the severity of variant consequence, as defined by Ensembl and based on genetic variant annotation and predicted effects (e.g., amino acid change and protein structure modification). The IMPACT rating categories are as follows: "modifier" (no evidence of impact), "low" (unlikely to have disruptive impact), "moderate" (non-disruptive but might have effect), and "high" (assumed to have disruptive impact). "Modifier" variants were not plotted but constituted a majority of variants in all samples. Variants categorized into "low," "moderate," or "high" are referred to as IMPACT mutations in the text. The variant count distribution was organized as a mean-centered mutation count per chromosome for samples within each comparison set (cell line versions and sublines). Variant overlap analysis was conducted using VCFtools and visualized using the *UpSet* (version 1.4.0) R package. Variants unique to each sample were plotted as proportions of the total mutations in that sample.

**Automatic generation of a genetic mutational signature.**   Unique cell line version mutations were input into *dNdScv* [98] to generate a mutational signature that could define the genetic heterogeneity within the PC9 cell line family (all cell line versions and sublines). In this analysis, genes are first annotated by type. *dNdScv* then uses a maximum-likelihood model to detect genes under positive selection (i.e., potential driver mutations). For each gene, a variety of models were utilized to identify genes that have substantially more nonsynonymous mutations than expected in each of the nonsynonymous mutation types, as compared to synonymous mutational load. These metrics were combined together to calculate a global *p*-value (see original publication [98] for details). We used this approach, with a maximum number of mutations per gene per sample = 5 (tool recommendation; limits a hypermutator phenotype), to determine genes with a global *p*-value <0.05 in all 8 members of the PC9 cell line family. The resulting gene signature set the baseline for the genetic heterogeneity for all cell populations (see gene list in Table B in S1 Text). We visualized the mutation data as a heatmap of these significant genes and cell line versions or sublines, colored by annotated mutation type (number of variants in each gene-population pair are not annotated in heatmap but are shown in adjacent bar charts for number of variants per gene and sample).

**Literature-curated, cancer-associated mutational signature.**   A cancer-associated gene list was established to supplement the predicted genetic heterogeneity signature from *dNdScv*. The gene list was created from the NIH Genetics Home Reference (GHR) key lung cancer genes (ghr.nlm.nih.gov/condition/lung-cancer#genes) and 2 additional publications of key mutations in lung cancer [2,59] (see S4A Fig for the gene names). Associated heatmaps were generated of this gene list for cell line versions and sublines, colored by annotated mutation type.

## RNA single-cell transcriptome sequencing

**Data acquisition.**   scRNA-seq libraries were prepared using the 10X Genomics gene expression kit (version 2, 3′ counting [99]; S5 Fig) and cell hashing [100]. Cells were prepared according to recommendations from the cell hashing protocol on the CITE-seq website (cite-seq.com/protocol). After cell preparation, 1 ng of 8 different cell hashing antibodies (TotalSeq-A025(1–8) anti-human Hashtag, BioLegend, San Diego, CA, USA) were used to label each of the 8 samples (labeling efficiency in S14 Fig). Hashed single-cell samples were combined in approximately similar proportions and "super-loaded" (aiming for approximately 20,000 cells, approximately 15,400 cells were obtained) onto the Chromium instrument. Cells were encapsulated according to manufacturer guidelines. Single-cell mRNA expression libraries were prepared according to manufacturer instructions. The leftover eluent of the mRNA expression library, containing the hashtag oligonucleotides (HTOs), was utilized to further size select the HTO library. The size-selected HTO library was PCR amplified to obtain higher-quality reads. Libraries were cleaned using SPRI beads (Beckman Coulter, Brea, CA, USA) and quantified

using a Bioanalyzer 2100 (Agilent). The libraries were sequenced using the NovaSeq 6000 with 150 bp paired-end reads targeting 50 M reads per hashed sample for the mRNA library and a spike-in fraction for the HTO library. RTA (version 2.4.11; Illumina) was used for base calling and MultiQC (version 1.7) for quality control. Gene counting, including alignment, filtering, barcode counting, and unique molecular identifier (UMI) counting, was performed using the *count* function in the 10X Genomics software *Cell Ranger* (version 3.0.2) with the GRCh38 (hg38) reference transcriptome (S5A Fig). We utilized *CITE-seq-Count* (github.com/Hoohm/ CITE-seq-Count) to count HTOs from the HTO library. We then used the *Demux* function in the R package Seurat [101] (satijalab.org/Seurat, version 3.2.2) to demultiplex the HTO and mRNA libraries and pair cells to their associated hashtag. Data were integrated into a count matrix with genes and cells, with HTO identity as a metadata tag.

**Data analysis.** After creating the demultiplexed, single-cell gene expression matrix, we removed cell multiplets using both cell hashtags (S15A Fig; at least 2 different hashtags detected with a single cell barcode) and automated doublet detection (*DoubletFinder* [102] version 3 with default parameters; S15B Fig). Additionally, poor-quality cells were removed based on a minimum cutoff of features (number of genes detected in each cell = 3,000) and count (number of RNA molecules detected within each cell = 15,000). These numbers were chosen subjectively but with respect to the overall sequencing depth in order to remove droplets with ambient RNA. Cells below these thresholds fell in isolated regions of the UMAP space and had a large degree of overlap with all other samples (S15C Fig). Feature selection was performed according to Seurat guidelines ($0.1 <$ average gene expression $< 8$, log variance-to-mean ratio $> 1$; 574 genes met criteria). Data were visualized using the UMAP dimensionality reduction algorithm, as implemented in the Seurat [101] R package. To facilitate comparisons across cell line versions and sublines, we performed the UMAP projections in the space of all 8 cell line versions and PC9-VU sublines (PC9-VU, PC9-MGH, PC9-BR1, DS3, DS6, DS7, DS8, and DS9). To quantify overlap of cells between transcriptomic features, we performed k-means clustering of the cell line versions (k = 3) and the sublines (k = 2) using the *NBClust* package, which also identified the optimal number of clusters based on 30 different methods. Differential expression analysis was performed between a single sample (cell line version or subline) and the rest of the PC9 cell line family members using the Wilcoxon rank sum test [103] (as implemented in the *FindMarkers* function in Seurat, default settings). DEGs (adjusted $p < 0.05$) were used for downstream analyses (see "Gene Ontology analysis" below). In addition to UMAP, t-distributed Stochastic Neighbor Embedding (t-SNE) and principal component analysis (PCA) were also performed, using the Seurat implementation.

## Copy number variation analysis

CNVs were inferred from scRNA-seq data using inferCNV (Trinity CTAT Project, github. com/broadinstitute/inferCNV). The single-cell transcriptome count matrix (see "RNA single-cell transcriptome sequencing: Data acquisition" above), an annotation file (pairing each cell to its corresponding PC9 cell line family member), and a gene order file (derived from the GRCh38/hg38 gtf file) were used to create an inferCNV object. Separate objects were created for cell line versions (no reference group; an average of the 3 versions was used, default setting) and sublines (PC9-VU reference group; S10 Fig). The inferCNV analysis was run with a cutoff of 0.1 (default for droplet-based experimental methods). Clustering was performed based on annotation file groups (i.e., cell line versions and sublines). Analysis settings were to denoise the dataset and use a hidden Markov model for CNV predictions. Heatmaps of relative expression values (to the reference group) were output by chromosome for all cells in the analysis. Red corresponds to increased expression (amplification) and blue to decreased expression (deletion).

### scRNA-seq functional interpretation analysis

The single-cell transcriptome count matrix (see "RNA single-cell transcriptome sequencing: Data acquisition" above) was scaled by multiplying counts by the median RNA molecules across all cells and dividing that number by the number of RNA molecules in each cell (as recommended). Gene signature files were obtained from the molecular signatures database (MSigDB). *Hallmark* gene sets (50 in total) were downloaded from MSigDB (gsea-msigdb.org/gsea/msigdb/genesets. jsp?collection=H). Both the scaled counts matrix and each of the hallmark gene sets were input into VISION [104] to identify gene signature scores for each cell-signature pair. Four hallmark gene sets (KRAS_SIGNALING_UP, KRAS_SIGNALING_DOWN, UV_RESPONSE_UP, and UV_RESPONSE_DOWN) were condensed into 2 (KRAS_SIGNALING and UV_RESPONSE) by VISION to leave 48 total gene signatures. Scores were compiled into a distribution and plotted by PC9 cell line family member for each gene set.

### RNA bulk transcriptome sequencing

**Data acquisition.** Total RNA was extracted using a Trizole extraction (ThermoFisher), according to the manufacturer's protocol. RNA-seq libraries were prepared using 200 ng of total RNA and the NEBNext rRNA Depletion Kit (New England Biolabs, Ipswich, MA, USA, Cat: E6310X), per manufacturer's instructions. The kit employs an RNaseH-based method to deplete both cytoplasmic (5S rRNA, 5.8S rRNA, 18S rRNA, and 28S rRNA) and mitochondrial ribosomal RNA (12S rRNA and 16S rRNA). The mRNA was enriched via poly-A-selection using oligoDT beads and then the RNA was thermally fragmented and converted to cDNA. The cDNA was adenylated for adaptor ligation and PCR amplified. The resulting libraries were quantified using a Qubit fluorometer (ThermoFisher), Bioanalyzer 2100 (Agilent) for library profile assessment, and qPCR (Kapa Biosciences, Cat: KK4622) to validate ligated material, according to the manufacturer's instructions. The libraries were sequenced using the NovaSeq 6000 with 150 bp paired-end reads. RTA (version 2.4.11, Illumina) was used for base calling and MultiQC (version 1.7) for quality control. Reads were aligned using STAR [105] (version 2.5.2b) with default parameters to the STAR hg38 reference genome. Gene counts were obtained using the feature-Counts [106] package (version 1.6.4) within the Subread package. The gene transfer format (GTF) file for the genes analyzed in the scRNA-seq data (provided by 10X Genomics and used in the *Cell Ranger* pipeline, generated from the hg38 reference transcriptome) was used to better facilitate internal comparison between scRNA-seq and bulk RNA-seq datasets.

**Data analysis.** RNA-seq data were analyzed using the DESeq2 [107] R package. Counts were transformed using the regularized logarithm normalization algorithm, as implemented in the *rlog* function of DESeq2. PCA was performed using the *prcomp* function in R and hierarchical clustering using the *hclust* R function with a Ward's minimum variance method. Data were visualized using the *ggplot2* R package.

### Gene ontology analysis

**Setup.** Genes associated with unique IMPACT mutations (classified as "low," "moderate," or "high" IMPACT scores; see "DNA bulk whole exome sequencing: Genomic mutational analysis") were identified for each comparison set (i.e., cell line versions or sublines). Additionally, DEGs from the scRNA-seq statistical comparisons for each comparison set were determined (see "RNA single-cell transcriptome sequencing: Data analysis" above). The 2 gene lists were independently subjected to a GO enrichment analysis using *EnrichR* [108] (version 2.1). Genes were compared to the ontology databases *GO Biological Process 2018* (BP), *GO Molecular Function 2018* (MF), and *GO Cellular Component 2018* (CC), which we refer to as GO "type" in the text.

**Correlation analysis.** GO terms significantly enriched in the IMPACT mutations ($p < 0.05$) and in DEGs ($p < 0.05$) were identified and stored independently as separate GO term lists for each PC9 cell line family member. For terms shared between the lists, we calculated $-\log_{10}(p)$ to rank terms based on statistical significance. Spearman correlation was calculated between the significant GO terms using the *ggpubr* R package (version 0.4.0), as long as a minimum of 5 significant terms were shared between the IMPACT and DEG GO term lists. Outlier GO terms ($-\log_{10}(p) > 10$) were excluded from the analysis (2 terms in PC9-BR1, 1 in PC9-MGH), in order to not unfairly skew correlation calculations.

**Semantic similarity analysis.** GO term lists for each PC9 cell line family member were further separated into GO types, which created GO term lists unique for each combination of sample (cell line version or subline), data modality (IMPACT mutations or DEGs), and GO type ("BP," "MF," or "CC"). For each sample, the mutation and DEG GO term lists associated with each GO type were compared using the semantic similarity metric from Wang and colleagues [64], as calculated in the *GOSemSim* package [109] (version 2.12.1) using the *goSim* function. This approach compares 2 individual GO terms using the underlying GO term network structure. Pairwise similarities were calculated on the lists of terms to generate similarity matrices for each sample. In order to avoid the dismissal of terms near any proposed statistical cutoff and ensure lists were of a minimum length, mutation and DEG GO term lists associated with each GO type for each sample were chosen randomly based on a modified *p*-value metric from the GO enrichment analysis. Specifically, terms were chosen from each list if a random number (between 0 and 1) was greater than the GO enrichment *p*-value. Semantic similarity distributions had a large skew, biased heavily toward lower values, primarily due to the size of the GO type graph network structure and, therefore, the "distance" between terms in the graph. To combat this issue, a maximum range of semantic similarity scores were chosen for each comparison (similar to but more robust than the "best max average" option provided in *GOSemSim*). The median of the top 1,000 scores and a 95% confidence interval were calculated for each sample-GO type comparison.

Semantic similarity scores were also correlated with the number of genes input into the GO enrichment analysis. To address this problem, random gene lists of the same lengths as IMPACT and DEG gene lists were chosen and input into the same process as experiment-derived gene lists in order to generate a simulated semantic similarity distribution. Depending on the length of the lists, these simulated distributions varied in the both the median and variance. Therefore, instead of comparing experimental and simulated distributions, experimental semantic similarity scores were normalized to the median + one standard deviation of the simulated score, for each sample. These relative GO semantic similarity score distributions are represented in plots. Importantly, the number of GO terms varied across GO types (according to *GOSemSim*; BP: 12,272, MF: 4,165, and CC: 1,740). We assume that GO types with more terms are more biologically significant, i.e., BP is more predictive than MF, followed by CC. Data were visualized using the *ggplot2* R package.

## In silico modeling of clonal fractional proliferation

**Birth-death population growth models.** Mathematical models of population growth dynamics were constructed using PySB [110], a Python-based kinetic modeling and simulation framework. We modeled cell proliferation as a simple birth-death process,

$$Cell_i \overset{k_{div,i}}{\rightarrow} Cell_i + Cell_i \tag{1}$$

$$Cell_i \overset{k_{dth,i}}{\rightarrow} \emptyset \tag{2}$$

where $i$ is an integer index, $k_{div,i}$ and $k_{dth,i}$ are division and death rate constants, respectively, for cell type $i$, and $\emptyset$ denotes cell death. Note that there is no state switching included in the model. Models with one cell type were used to compare against experimental cFP data for the majority of the PC9-VU sublines (DS1, DS3, DS4, DS6, DS7, and DS9), while a two-cell-type model was used in one case (DS8).

**Stochastic simulations and in silico DIP rate distributions.** All model simulations were run using the stochastic simulation algorithm (SSA) [65,88], as implemented in BioNetGen [111] (invoked from within PySB), to capture the effects of random fluctuations in division and death on cell population proliferation. We performed in silico cFP assays, where numerous single cells (run as independent simulations) were grown into colonies of variable size over 8 days of simulated time using the SSA and fixed rate constants for division and death ($k_{div}$ = 0.04·ln(2) h$^{-1}$, $k_{dth}$ = 0.005·ln(2) h$^{-1}$; Table C in S1 Text), based on vehicle-control proliferation data (Fig 1C, dashed lines; in the case of 2 cell types, both types were assumed to grow at the same rate outside drug). We ran as many simulations as there were experimental cFP trajectories for the PC9-VU subline being compared against (Fig 5A and 5E and S12A Fig). Drug treatment was then modeled by changing the rate constants for division and death (for 2 cell types, each was assumed to proliferate at different rates in drug; Table C in S1 Text) and running for the additional days of simulated time corresponding to each subline experiment. Simulated time courses were plotted at the same time points as in the corresponding experimental cFP assays for direct comparison. In silico DIP rates were obtained by taking $\log_2$ of the total cell counts and calculating the slope of a linear fit to the time course from the time of drug addition to the end of the simulation using the SciPy [112] *linregress* function. DIP rates for all in silico colonies were compiled into a distribution and compared to the corresponding experimental cFP distribution using the AD test [66]. The *p*-value for each simulation result was bootstrapped based on 100 resamples of the experimental distribution. For DS8, a two-state model was used. All aspects of the model were the same, except drug treatment was modeled by 2 sets of division and death rate constants. These simulations were constrained by the approximate DIP rate ranges for both DIP rate distribution modes (Fig 1D).

**Parameter scans.** We repeated the simulation procedure above over ranges of physiologically plausible division and death rate constant values (Table C in S1 Text). For each parameter set (either 1 or 2 pairs of postdrug division/death rate constants, depending on subline), AD tests for simulated versus experimental DIP rate distributions were performed. To account for variabilities in individual comparisons, only simulations with a mean *p*-value minus one standard deviation from the bootstrapped result were kept. Any simulation with <p>−sdev(p) > 0.05 (i.e., we cannot reject the null hypothesis that the DIP rates are drawn from the same distribution) were plotted in a heatmap using the *ggplot2* R package. Thus, all colored elements in the heatmap represent sets of division/death rate constants that produce DIP rate distributions statistically indistinguishable from the experimental distributions obtained from cFP assays. Note that the scan for the two-cell-type model (DS8) was performed in 4 dimensions (2 division and 2 death rate constants), but results were plotted in 2 dimensions for visual simplicity.

## Supporting information

**S1 Text. Extended discussions regarding genetic, epigenetic, and stochastic sources of tumor heterogeneity.** Four subsections are included, discussing (i) descriptions of the genetic, epigenetic, and stochastic levels of heterogeneity believed to coexist within tumors, (ii) the fundamental relationship between transcriptomics and epigenetics, (iii) the rationale behind the genetic-to-epigenetic correlation metric utilized in this work, and (iv) 2 hypotheses regarding the origins of the DS8 genetic mutant subline. Three tables are also included containing (A) a

glossary of terms, (B) a list of genes associated with mutation heatmaps in Figs 3D and 4D, and (C) rate parameters used for stochastic simulations.
(PDF)

**S1 Fig. The PC9 cell line family.** (A) Identification of canonical EGFR-ex19del in PC9 cell line family members. A screenshot from the IGV is shown. Red corresponds to potential deletions and blue to potential insertions. The data underlying this image can be found in the Sequence Read Archive (ncbi.nlm.nih.gov/sra) at accession #PRJNA632351. (B) PC9 cell line family tree. Two versions the PC9 cell line were maintained separately in culture at 2 different institutions (VU and MGH). A resistant cell line (PC9-BR1) was derived from PC9-VU by dose escalation in the EGFRi afatinib. Several DS were also single-cell isolated from PC9-VU. Colors are consistent with data visualizations in main and supplementary figures. DS, discrete subline; EGFR, epidermal growth factor receptor; EGFRi, EGFR inhibitor; IGV, Integrative Genomics Viewer; MGH, Massachusetts General Hospital; VU, Vanderbilt University.
(TIF)

**S2 Fig. cFP assays for PC9 cell line versions.** (A) PC9-MGH treated with erlotinib. (B) PC9-BR1 treated with erlotinib. All trajectories in A and B are normalized to approximately 72 h postdrug treatment. (C) PC9-VU treated with erlotinib. Trajectories for the parental line (gray) and the discrete sublines (colors) are plotted together for comparison. All subline trajectories, except DS8, are normalized to approximately 125 h post-erlotinib treatment; DS8 was normalized to the time of treatment because it was resistant and reached confluency during the course of the experiment. For the sublines, means of time point replicates are plotted. In all cases, number of colonies (n) are noted within the plots. The data underlying this figure can be found in github.com/QuLab-VU/GES_2021. cFP, clonal fractional proliferation.
(TIF)

**S3 Fig. Metrics of WES data.** (A) Total number of mutations identified through variant calling compared to hg38 reference genome. Mutations are separated into substitutions, specifically SNPs and InDels. (B) Sequencing quality metrics for the PC9 cell line family (considered together as one group). DP is a measure of sequence coverage; MQ details how well the sequencing reads are mapped to the reference genome; QUAL is a score developed for Phred base calling that measures the confidence in called variants based on sequencing error probabilities; variant count is a reflection of the variants per site identified over small sections (windows) of the reference genome. (C–E) Quantified Venn diagram (i.e., UpSet plot) of unique, and intersections of, mutations in (C) cell line versions, (D) PC9-VU sublines, and (E) PC9-VU sublines and parental. The data underlying this figure can be found in github.com/QuLab-VU/GES_2021. InDel, insertion/deletion; SNP, single nucleotide polymorphism; WES, whole exome sequencing.
(TIF)

**S4 Fig. Additional mutation analyses across cell line versions and sublines.** (A) Mutational differences between PC9 cell line family members for a literature-curated set of cancer-associated genes implicated in lung cancer (see Materials and methods). Heatmap elements are colored based on type of mutation. (B) Mutation class pie charts. The data underlying this figure can be found in github.com/QuLab-VU/GES_2021. Indel, insertion/deletion; SNV, single nucleotide variant.
(TIF)

**S5 Fig. Metrics for scRNA-seq comparisons.** (A) Cell Ranger (support.10xgenomics.com/single-cell-gene-expression/software) output file detailing metrics of sequencing run (quality,

mapping, barcode identification, etc.). The data underlying this image can be found in the Gene Expression Omnibus (ncbi.nlm.nih.gov/geo) data repository at accession #GSE150084. (B) Feature identification for genes that transcriptomically differentiate PC9 cell line family members. Variable genes are projected on a plot of dispersion vs. average gene expression and genes that pass a feature selection threshold are shown in red (0.1<average gene expression<8, log variance-to-mean ratio>1; 574 genes). The data underlying this plot can be found in github.com/QuLab-VU/GES_2021. scRNA-seq, single-cell RNA sequencing; UMI, unique molecular identifier.
(TIF)

**S6 Fig. Clustering and alternative visualizations of scRNA-seq data.** (A) Clustering of cell line versions. Number of clusters (3) was defined based on majority rule from a consensus of 30 indices. Ward's minimum variance method was used. (B) Quantification of cluster fraction by cell line version. (C) Same as A but for sublines. Two clusters were found to be the consensus. (D) Same as B but for sublines. (E) PCA visualization of single-cell transcriptomes. (F) t-SNE visualization of single-cell transcriptomes. The data underlying this figure can be found in github.com/QuLab-VU/GES_2021. PCA, principal component analysis; scRNA-seq, single-cell RNA sequencing; t-SNE, t-distributed Stochastic Neighbor Embedding.
(TIF)

**S7 Fig. Bulk RNA-seq data.** (A) PCA of single-replicate normalized RNA-seq count data. (B) Hierarchical clustering of RNA-seq normalized count data. Clustering was performed on the pairwise Euclidian distance matrix created from the relative log transformed gene counts using the Ward's minimum variance method. The data underlying this figure can be found in github.com/QuLab-VU/GES_2021. PCA, principal component analysis; RNA-seq, RNA sequencing.
(TIF)

**S8 Fig. VISION transcriptome functional interpretation analysis.** Single-cell gene expression matrix and MSigDB hallmark gene signatures were input to create a signature score for each cell. Scores were totaled for each population across each hallmark and plotted as a density distribution. All 50 hallmark signatures were sampled. Note that "KRAS signaling" and "UV response" had hallmark signatures for both up- and down-regulated. We condensed these 4 signatures into 2, leaving 48 hallmark signatures total. The data underlying this figure can be found in github.com/QuLab-VU/GES_2021. MSigDB, molecular signatures database.
(TIF)

**S9 Fig. GO semantic similarity scores for each GO ontology type.** Significantly enriched GO terms for each data modality pair (IMPACT mutations and DEGs) were compared for each cell line family member for each GO type (BP, MF, and CC). The top 1,000 similarity scores within each pair were compiled into a distribution to calculate a median (white circle) and 95% confidence interval (error bars). Scores are plotted relative to a baseline, defined as the median + one standard deviation of simulated distributions (dashed lines). Simulated score distributions were calculated based on random gene lists of identical lengths to the experimental gene lists (see Materials and methods). The data underlying this figure can be found in github.com/QuLab-VU/GES_2021. BP, Biological Process; CC, Cellular Component; DEG, differentially expressed gene; GO, Gene Ontology; MF, Molecular Function.
(TIF)

**S10 Fig. Baseline expression values across chromosomes for PC9-VU parental.** Values are plotted as a heatmap. All PC9-VU sublines were compared against this baseline in the CNV

analysis. The data underlying this figure can be found in github.com/QuLab-VU/GES_2021. CNV, copy number variant.
(TIF)

**S11 Fig. Potential explanations for cell state heterogeneity in DS8.** (A) Multiple genetic states hypothesis. In this scenario, a genetic resistance mutation was acquired after the DS8 subline was established. Assuming the mutant state does not outgrow the original genetic state (i.e., a "selective sweep"), both genetic states should coexist within the subline. (B) Single genetic state hypothesis. In this scenario, a genetic resistance mutation emerged within the PC9-VU parental population and a cell containing that mutation was isolated to establish the DS8 subline. To explain our single-cell transcriptomics data, we hypothesize that cell–cell interactions between mutant and PC9-VU cells increase the death rate for mutant cells, making them a small proportion (<2%) of the total PC9-VU population.
(TIF)

**S12 Fig. Comparison of experimental and simulated cFP assays for PC9-VU sublines.** (A) Experimental cFP time courses for 4 PC9-VU sublines (DS1, DS6, DS7, and DS9) in response to 3 μM erlotinib (same data used to generate DIP rate distributions in Fig 2D of the main text). Each trace corresponds to a single colony, normalized to 72 h postdrug treatment. Only colonies with cell counts greater than 50 at the time of treatment were kept; n represents the number of colony traces for each subline. (B) Simulated cFP time courses generated using division and death rate constants that closely reproduce the experimental time courses in A. Trajectories are normalized to the time at which the simulated drug treatment was initiated and simulated cell counts are plotted only at experimental time points. Although the same number of simulations were initiated as the number of colonies (n) in the corresponding experiment (see panel A), only simulated colonies with cell counts >50 at the time of simulated drug treatment are shown. (C) Comparison of experimental and simulated DIP rate distributions calculated from time courses in A and B. Distributions are compared statistically using the AD test (see Materials and methods). Bootstrapped $p$-values are shown (mean and standard deviation). Dashed black line signifies zero DIP rate, for visual orientation. (D) Parameter scan of division and death rate constants for the 4 sublines in A–C. For each pair of rate constants, we ran model simulations (same number as corresponding subline), calculated DIP rates, compiled them into a distribution, and then statistically compared against the corresponding experimental DIP rate distribution using the AD test (bootstrapped). All $p < 0.05$ are colored white, indicating lack of statistical correspondence to experiment. × denotes a division and death rate constant used in B. The data underlying this figure can be found in github.com/QuLab-VU/GES_2021. AD, Anderson–Darling; cFP, clonal fractional proliferation; DIP, drug-induced proliferation.
(TIF)

**S13 Fig. Genomic relatedness between PC9 cell line family members.** (A) PCA of PC9 genotypes. Using a subset of SNPs in approximate linkage equilibrium, a genetic covariance matrix was calculated. The covariance matrix was converted to a correlation matrix to achieve appropriate scaling and PCA was run to identify SNP eigenvectors (loadings of the principal components). PC9 cell line family members are plotted along the principal component axes. (B) Hierarchical clustering of PC9 genotypes. Using an identity-by-state analysis, a matrix of genome-wide pairwise identities was calculated. Hierarchical clustering was performed on these identities to determine sample relatedness. The data underlying this figure can be found in github.com/QuLab-VU/GES_2021. PCA, principal component analysis; SNP, single nucleotide polymorphism.
(TIF)

**S14 Fig. Sample identification in "hashed" PC9 cell line family members.** Proportional representation of cell populations with each of 8 specific "hashtag" antibodies, based on the HTO expression level. Each sample has a single corresponding HTO, while a minority of the HTO reads were unmapped. The data underlying this figure can be found in github.com/QuLab-VU/GES_2021. HTO, hashtag oligonucleotide.
(TIF)

**S15 Fig. scRNA-seq quality control analyses.** (A) Cell hashing allowed for detection of cell multiplets (1 droplet with more than 1 cell) because multiple HTOs would be detected for a single cell barcode (i.e., droplet). Cells were segregated into singlets and doublets (i.e., multiplets). All detected cell transcriptomes were visualized using UMAP, noting singlets and doublets. (B) Automated DD was performed on the detected cell transcriptomes. Doublets were predicted and noted on the same UMAP visualization as in A. (C) Cells were scored based on number of features (e.g., genes) and count of detected RNA molecules. Cells with scores below a specified threshold (see Materials and methods) were classified as "poor" quality. Quality of each cell was noted on the same UMAP visualizations as in A and B. A total of 7,892 cells passed singlet and quality control thresholding. The data underlying this figure can be found in github.com/QuLab-VU/GES_2021. DD, doublet detection; HTO, hashtag oligonucleotide; scRNA-seq, single-cell RNA sequencing; UMAP, Uniform Manifold Approximation and Projection.
(TIF)

# Acknowledgments

This work is dedicated to the memory of our friend and colleague Melaine N. Sebastian. We thank Jing Hao for reagent acquisition and Tony Capra, Bishal Paudel, Christian Meyer, Sarah Maddox Groves, Carlos Lopez, Alissa Weaver, John McLean, Maizie Zhou, and Ken Lau for useful discussions.

Sequencing studies were performed by the Vanderbilt Technologies for Advanced Genomics (VANTAGE, Vanderbilt University Medical Center) core, an institutionally supported core facility with help from Angela Jones, Karen Beeri, Jamie Roberson, Latha Raju, and Matthew Scholz. Sorting of labeled cells and single-cell seeding for cFP assays were performed with oversight from the Flow Cytometry Shared Resource (Vanderbilt University Medical Center). Drug changes on 384-well plates for cFP assays were performed at the Vanderbilt High-Throughput Screening (HTS) facility. Data processing and model simulations were performed using the computational resources available at the Advanced Computing Center for Research and Education (ACCRE) at Vanderbilt University.

# Author Contributions

**Conceptualization:** Corey E. Hayford, Darren R. Tyson, Vito Quaranta, Leonard A. Harris.

**Data curation:** Corey E. Hayford, C. Jack Robbins, III, Peter L. Frick.

**Formal analysis:** Corey E. Hayford, C. Jack Robbins, III, Leonard A. Harris.

**Funding acquisition:** Darren R. Tyson, Vito Quaranta.

**Investigation:** Corey E. Hayford, C. Jack Robbins, III, Leonard A. Harris.

**Methodology:** Corey E. Hayford, Darren R. Tyson, Peter L. Frick, Leonard A. Harris.

**Resources:** Vito Quaranta.

**Supervision:** Vito Quaranta, Leonard A. Harris.

**Writing – original draft:** Corey E. Hayford, Vito Quaranta, Leonard A. Harris.

**Writing – review & editing:** Corey E. Hayford, Darren R. Tyson, Vito Quaranta, Leonard A. Harris.

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
