## [Editor Report · Decision Letter 0]

29 May 2020

Dear Dr Harris, 

Thank you for submitting your (somewhat revised) manuscript entitled "A unifying framework disentangles genetic, epigenetic, and stochastic sources of drug-response variability in an in vitro model of tumor heterogeneity" for consideration as a Research Article by PLOS Biology.

Your manuscript has now been evaluated by the PLOS Biology editorial staff, and I'm writing to let you know that we would like to send your submission out for external peer review.

Please re-submit your manuscript within two working days, i.e. by Jun 02 2020 11:59PM.

Kind regards,

Roli Roberts

Senior Editor

PLOS Biology

---

## [Decision Letter · Decision Letter 1]

23 Aug 2020

Dear Dr Harris,

Thank you very much for submitting your manuscript "A unifying framework disentangles genetic, epigenetic, and stochastic sources of drug-response variability in an in vitro model of tumor heterogeneity" for consideration as a Research Article at PLOS Biology. Your manuscript has been evaluated by the PLOS Biology editors, an Academic Editor with relevant expertise, and by three independent reviewers. Please accept my apologies for the unusual amount of time that the review process has taken during these challenging times.

The reviews of your manuscript are appended below. You will see that the reviewers find the work potentially interesting. However, based on their specific comments and following discussion with the academic editor, I regret that we cannot accept the current version of the manuscript for publication. We remain interested in your study and we would be willing to consider resubmission of a comprehensively revised version that thoroughly addresses all the reviewers' comments. We cannot make any decision about publication until we have seen the revised manuscript and your response to the reviewers' comments. Your revised manuscript would be sent for further evaluation by the reviewers.

IMPORTANT: Having discussed the reviews with the academic editor, it is clear that the reviewers' comments are strikingly concordant. In each case they are impressed by the scale of your study, and think that is potentially appropriate for publication in our journal. HOWEVER, all of the reviewers request that you considerably improve the structure and presentation of your study, shortening the manuscript and improving the clarity and focus. There are also some requests for additional analyses that will enhance the utility and appeal of the paper for our readership. We encourage you to take these comments to heart and to carry out a thorough and conscientious re-working of the manuscript.

We appreciate that these requests represent a great deal of extra work, and we are willing to relax our standard revision time to allow you six months to revise your manuscript.We expect to receive your revised manuscript within 6 months.

**IMPORTANT - SUBMITTING YOUR REVISION**

*Resubmission Checklist*

*Published Peer Review*

*PLOS Data Policy*

*Blot and Gel Data Policy*

Sincerely,

Roli Roberts

Senior Editor,

rroberts@plos.org,

PLOS Biology

REVIEWERS' COMMENTS:

Reviewer #1: 

 In their manuscript, Hayford and coworkers apply a framework relating genetics, epigenetics, and stochasticity to derivatives of the non-small-cell lung cancer line, PC9. The framework is Waddington's epigenetic landscape, where the tent strings underneath the landscape are pulled to varying extents depending on mutations or other perturbations (PMID 29590606). Cell-to-cell heterogeneity rocks the landscape and opens the possibility of a cell escaping one attractor and entering another. The authors deeply describe the molecular and phenotypic characteristics of PC9 subclones along with polyclonal lines from different laboratories, performing exome sequencing, RNA-seq at the population and single-cell level, and fractional proliferation assays The goal is to reconcile mutations, single-cell fluctuations in gene-expression state, and drug-induced proliferation rates by using the framework. It is difficult to say whether they accomplished that goal, but the multi-omics data provide a lot of observations that could be followed up on in the future.

 The major issue I have with the manuscript is bloat. There are three types of experiments here—1) exome sequencing, 2) single-cell/bulk transcriptomics, 3) time-lapse recordings of inhibitor-treated cells—and associated analyses. Given the scope of the work and the findings, 87 pages is not impressive but agonizing. The Waddingtonian landscape does not require two pages to introduce, nor do we need a dozen examples of how noise can arise in biology. The experimental results in Figure 6 are a carbon copy of Figure 4 expanded by one PC9 subclone, artificially creating a density to the work that it does not deserve. The Supplementary Note is not a note at all but instead another dozen pages of text that editorializes on the model, gives an introductory primer on the different types of mutational events, and speculates on the role of stochasticity. The only text that really belongs there is the description of the growth model, as it is essential to understand the results shown in the main text. The authors should go back and look critically at every text passage, supplementary figure subpanel, and yet-another analysis toward justifying why each element is absolutely essential for their message.

 Quite possibly, there are some interesting results here. If the UMAPs of Fig. 3G, 4G, and 6C are reassembled into a single graph, one sees an outlier transcriptional state in PC9-VU, which is recapitulated almost exactly in the DS9 clone—could this clone be thought of as a cancer stem cell-like founder for PC9s, with an epigenetic landscape that is shallow (or noisy) enough to populate the other basins? Reciprocally, the DS8 clone appears to be trapped in the outlier state—does the increased mutational burden mechanistically explain the altered landscape? Lip service to a mutated ABC transporter and a silent mutation in RELN is insufficient. Real biological connections that are plausibly mechanistic would allow the authors to look for related connections in other NSCLC datasets.

 The quantity of comments relates directly to the volume of material provided.

Major points

1. Novelty and utility of the genetic-epigenetic-stochastic (G/E/S) framework. The manuscript pseudo-presents a set of ideas at the beginning as if they have never been described before. In fact, prior work by the authors follows almost the exact same organizing principles, with "genotype" substituted for "drug treatment" (PMID 29590606). It is fine to build off past examples—simply cite the best and most-informative examples and get on with the new results here. Second, while I do not doubt that the G/E/S framework provides a straightforward physics-based analogy, I question whether it provides more than a cartooned explanation of the findings. The framework has no predictive value, and the sketches are so flexible that it appears to be consistent with whatever data are at hand. Is there anything that could falsify the G/E/S framework? A theory that explains everything explains nothing.

2. Lack of clarity on the meaning of "Unknown" exonic mutations. Amidst all the text in the Supplementary Note defining SNPs, indels, and missense mutations, I can find no explanation for how exome sequencing can yield an "Unknown" mutation. The only way I can fathom is if the capture beads pulled down collateral bits of intronic and extragenic sequence, but then those mutations should be omitted from the data reporting as they are not in exons. Most of the mutations reported in Fig. 3D and S10D fall into the "Unknown" category, so it is critical to know what these are.

3. Lack of consistency in exome sequencing results. There are also inconsistencies in the exome data that were not acknowledged. For example, ZNF717 is reported to be mutated in three of the four clones in Fig. 4D. One would expect that frequency of mutation to be picked up in the PC9-VU bulk population, but no ZNF717 mutation is reported in Fig. 3D. Furthermore, only two ZNF717 mutations are reported among the five clones shown in Fig. S10B, which is supposed to be a one-clone expansion of the data in Fig. 4D.

4. Copy number alterations are ignored. The experimental implementation of the G/E/S framework focuses entirely on mutations and does not consider the impact of chromosome stability and copy-number alterations. In a larger, thematically-similar study of HeLa cells (PMID 30778230), copy-number alterations were a driving force for heterogeneity among derivatives of the line. The authors could address this concern without any further experiments by applying inferCNV to their bulk RNA-seq data on each PC9 derivative and clone. CCLE RNA-seq data on PC9s could serve as a "normal" starting reference, assuming that CCLE obtained their PC9s directly from the original source.

5. Equating transcriptional and epigenetic states. The authors acknowledge that epigenetics is more conventionally viewed in terms of histone and DNA modifications. My issue is less with the measurement type and more with the implied lifetime of the state if it is framed epigenetically. For a state to be epigenetic, there should be some degree of heritability before the state "relaxes" over several generations. But, some transcriptional states just randomly switch because of noise or other reasons that have nothing to do with epigenetics (see https://doi.org/10.1101/379016 for a nice description of the distinction between the two). The outlier state in PC9-VU and DS9 would be a prime candidate for assessing lifetime, if there were surface markers that could be sorted for (PMID: 28607484).

6. Proof-by-intimidation reporting of the omics data. Whereas the cFP and DIP experiments are generally clear, well described, and informative, the omics data is presented in manner that overwhelms the reader more than it empowers them. Taking Fig. 3D-E as a case in point, the text is illegible at anything approaching a normal page, the shading that redundantly encodes the bar graph to the right is imperceptible even on a retina display, and the cancer genes list is effectively devoid of information (while being crammed with gridded lines and microscopic font). I would recommend a phased presentation, where the most-critical results are presented in main figures, and the gory details are presented ONCE in the supplementary figures (i.e., Figure S10A-D and removing Fig. 4A, D-F).

7. Excluding trivial clone-to-clone differences. As understood, the PC9-VU clones were isolated from a polyclonal population of PC7-VU cells ectopically expressing H2B-mCherry. The abundance of the H2B marker could readily change the epigenetic state of cells, as mCherry is much larger than H2B, and there are many unpublished anecdotes of H2B-mCherry cells behaving oddly, possibly the result of artifactually opening chromatin that should normally be closed. At the minimum, the authors should assess the relative abundance of H2B-mCherry in the different clones and confirm that the overall expression of the label is not dramatically different among clones.

Minor points

1. The "semantic similarity" is possibly a good way to bring together the mutational data with the transcriptomic data. However, there must be some type of interval estimate on the score and the polyclonal baseline used for comparison. Stability of the scores could be estimated by crossvalidation.

2. I can understand the motivation for "withholding" the DS8 clone in the presentation, but it really exacerbates the bloat of the manuscript. Also, in the presentation of the first four clones, it is not clear why DS9 and DS3 were selected, as they are very similar in their response to EGFR inhibition. Perhaps it is better to frame those two clones as a "within group" comparison to estimate how different the transcriptional states can be and yet yield similar drug-induced responses.

3. The formulation of the parameter inference of the stochastic DIP models is generally well done. However, I am not clear on why the division rates appear to flat-line at a division rate of ~0.045. I hope that was not simply the maximum division rate tested. Also, it is inappropriate to use large p values as a metric of goodness of fit. Much better would be to report the K-S statistic and shade to indicate the minimum value(s) that are deemed of interest.

4. This is not entirely the fault of the authors, but I find the V3 output of the GeneRanger software annoying for GEO uploads. The barcodes file precludes reviewers from quickly spot-checking the raw data against the figures. If the gene names could be exported along with the unique barcodes, that would be preferred.

Typographical points 

1. Page 5: The reference to Harris et al. should only refer to 89.

2. Page 7: The statement "The result above illustrate…" is inaccurate. The only thing that can be said is that the mutations are associated with transcriptional differences.

3. Page 11: The statement "However, individual basins are not discernable.." is false. The outlier basin is clearly visible in Fig. 3G.

4. Figure S8: There is no need for a legend here. Just label the data points.

5. Supplemental Note, Page 12. "reduce" should be "reduced".

Reviewer #2:

[identifies himself as Aaron S Meyer]

Hayford et al. provide a framework to deconvolve drug response variability into genetic, epigenetic, and stochastic sources in an in vitro heterogeneous tumor model. They explore the genetic and epigenetic differences that explain drug response variability using mutational impact, single-cell differential gene expression, and semantic similarity of gene ontology analysis across 3 versions of the PC9 NSCLC cell line and 8 sublines of PC9-VU. The authors conclude that the cell line versions, as well as single cell-derived sublines differ at genetic and transcriptomic levels. They argue that their framework could be employed to account for all levels of heterogeneity when evaluating cancer treatment.

Overall, the manuscript presents a comprehensive analysis of heterogeneity within a cell line model. It presents a valuable analysis that would be of interest to the drug response and tumor heterogeneity communities. Before it is suitable for publication, however, there are some major concerns outlined below that must be resolved. I expect the authors can fully address these concerns with text changes and some additional computational analysis.

Major concerns:

* I appreciate that the authors have aimed to be comprehensive in their definitions and discussion. However, parts of the paper seem very wordy. As one example, genetic, epigenetic, and stochastic heterogeneity are defined within separate sections, and then again defined within a supplementary table. If there are suitable references the paper can use to explain these definitions, this could help to focus the paper on the new results presented here. It would also make the manuscript more accessible overall.

* I understand the comparison to the ideas proposed by Waddington but think this need (1) be clarified, and (2) its ultimate usefulness highlighted. The Waddington landscape model specifically describes a developmental process, and how barriers between states might increase as development progresses. Where is the analogous process here? In what way can one use the attractor model as a testable hypothesis, or is this a tautological idea?

* The authors must at least acknowledge that heterogeneity can arise from environmental and pharmacologic differences. There is extensive literature on the contribution of these factors, even within simple tissue culture systems.

* The tense throughout the manuscript is inconsistent which leads to confusion. The authors seem to start by referring to their experiments in the past, and observations in the present, which reads well. However, this changes a few times throughout the text.

* Semantic similarity is an interesting idea for demonstrating conserved molecular programs. Can the semantic similarity score be evaluated compared to a meaningful null model for significance? For example, randomized gene sets of the same size from those genes expressed in PC9? Maybe it can also be shown that the similarity between types of molecular measurements is higher within a cell line (e.g. BR8) as compared to between lines? I am concerned that GO enrichment might simply be a reflection of the genes expressed in this cell line, and that semantic similarity simply scales with the number of impacted genes.

* The author's conclusions from their SSA model are overreaching and inconsistent. I agree that they can conclude their DIP rate measurements are consistent with the SSA model but do not agree that they can say much more. It is unclear to what extent their measurements are powered to reveal other forms of heterogeneity if present. Second, they are using one measure of phenotypic heterogeneity, when "epigenetic" heterogeneity is a much more encompassing term (e.g. drug response to other compounds). Third, their parameter uncertainty is simply that, and not evidence that the cells themselves fluctuate.

Minor comments:

* When comparing the SSA model to DIP measurements, the Anderson-Darling test should be more sensitive to differences as compared to the Kolmogorov-Smirnov test. Also, bootstrap resampling is necessary to prevent bias in the test statistic. This is summarized well here: https://asaip.psu.edu/articles/beware-the-kolmogorov-smirnov-test/

* I would not call PC9 a prototypical or archetypal example of lung cancer. It is a commonly used model, but like any cell line model (or any one tumor) presents a limited view of all lung cancer.

* There are a couple of places where there is the note: "(see below)". It is not clear what the authors are referring to.

* In figure 2B, the y-axis is density—is the magnitude of this quantity meaningful (e.g. in units of cell number)? If not, it would be better to normalize the distributions such that the y-axis would be 0-1.

* In figure 3, it is not clear to which plot "CV=12.84" belongs. Same in figure 4.

Reviewer #3:

Hayford et al tackle the issue of disentangling the genetic, epigenetic and stochastic sources of heterogeneity in cancer response to treatment. The authors tackle this problem with a combined approach of theoretical and mathematical modeling with genomics, single cell transcriptomic and experiments on drug response in three cancer cell lines and several monoclonal sublines derived from one of these three lines. This is a very important problem which has relevant repercussion on drug design and development of therapeutic strategies in cancer. The system and the approached used are clever and effective and the results are interesting. However, I think the authors could have done a better job in analyzing the large amount of data they produced, in particular the transcriptomic data, which could provide a much more detailed and informative characterization of the epigenetic landscape of the system. Moreover, the manuscript is massive, highly redundant and difficult to read because the information is poorly organized and scattered through main text, methods and supplementary note (the pdf is also missing line numbers which makes it difficult to review). Most importantly, the results can be presented in a much more concise and effective way. As the most striking example, almost all the data reported in figure 4 is repeated in figure 6 and supplementary figure S10 (even with different color codes), with the addition of another subline, making figure 4 B C D E F G H and I panels completely redundant. Also, in another example, the authors separate the sublines in different subplots in figure 2C when Figure S4 shows that sublines can be put in the same plot which is more concise and makes it is easier to compare between them. A profound reorganization of the figures and of the results section is necessary before publication. 

My major and minor points are detailed below:

Major points

1- Single cell RNA seq data is a tremendous resource to characterize the state of a cell but here we are left with very little information; the authors should extract more interpretable information from their data: For example, why don't they start from UMAP including all their data like the one they have in supplementary fig S7 i.e. both lines and sublines instead of splitting their analysis into three in the main text? Further, UMAP is a powerful visualization technique but it also lacks interpretability without integration or further analyses to characterize what the UMAP dimension 1 and 2 represent biologically. What are the molecular signatures and biological processes that separate the different clusters in the UMAP? What can we learn from these signatures about the epigenetic landscape? Which basin(s) of attraction - or cell state(s) is (are) i.e. associated with drug resistance? The authors could for example use the following tool to improve interpretability of their analysis: https://github.com/YosefLab/VISION. 

2- The GO semantic similarity analysis between the genetic and epigenetic level is potentially interesting but we lack information about the results and interpretation of this analysis. For example, the authors provide just an example of the analysis for subline DS8 in supplementary table S2 but they only provide the GO identifiers and not the names making any attempt to understand and interpreting the analysis very hard. What are the biological processes and molecular functions that drive the semantic similarities between the genetic and epigenetic level of the cell lines? Are those processes or function shared among some of the lines or sublines? 

3 - In addition if I understand correctly, the authors only compare each subline against the other sublines both at the genetic and RNA seq level, but wouldn't be interesting to compare them also against the line from which they are derived?

4- Why haven't the authors performed single cell RNA seq of DS1?

5 - It would be nice if the authors could confidently conclude that the BR1 lines and the subline DS8 which are both resistant to the drug, achieve resistance through two distinct genetic and/or epigenetic cell states. However From figure 4 G, (or figure 6 C) it seems to me that a minority of the DS8 cells occupy the same epigenetic state of BR1. It is therefore difficult to conclude that the resistance in DS8 is not driven by those cells that are close to BR1, also considering that the bimodal distribution in the DIP rate would suggest that it is a minority of cells that drive the resistance. Maybe the authors could sort and profile mutations and RNA of the high DIP rate and low DIP rate cells separately and look at where those cell maps in the genetic and epigenetic landscape?

6 - As explained above, authors should try to present the analyses of all lines and sublines combined. It doesn't make sense to repeat the analysis and presentation of the sublines data before and after adding the DS8, it's too redundant. Then, they can easily discuss the different nature of the DS8 with respect to the other sublines referring to the same plots.

Minor points

Main text

- First part of results i.e. the lengthy description of the three levels is actually more suitable for introduction and I'd suggest to shorten it to improve readability, or alternatively stick it to the supplementary note. 

- At page 7, when describing Fig 3 G i.e. the UMAP: 'we see a clear separation of the features for the cell line version in this space' this is confusing as features has been just used to indicate genes/transcript while describing the feature selection. 

I guess what the authors meant is: 'we see a clear separation of the cell lines in the UMAP gene expression space' or something on these lines. The same applies to the similar sentence at page 9. 

- At page 7: 'However most genetic mutations do not result in an altered expression of that gene'. Which gene?

- Why don't you describe IMPACT score just in the methods the first time you use it instead of just mentioning in the methods and then describe it in the supplementary note?

- Also check IMPACT is always used in upper case (I have seen at least one instance of lower case impact at page 16)

- You mention you didn't use the default covariates of dNdScv because they are not available for hg38, but you can't your own covariate? There are lots of possible variables (i.e. chromatin state) that impact mutation rate and possibly confound the analyses. 

- Cell cycle score: no details are provided on how this was computed.

- Feature selection: why filter out highly expressed genes? 

- K means clustering: is it performed on the whole expression matrix or just on the UMAP projection? Also, why k=3? what happens with K=4? Did you try a silhouette analysis to find the optimal number of clusters ?

- In silico modeling. Authors model all sublines with one set of division and death parameters apart from DS8 for which you used two sets. Did they try some sort of model comparison to determine if the sublines are better modeled with a single set of more than one set or they just decided by looking at the distribution of the DIP rate? From the RNA-seq epigenetic landscape it looks like the other lines might also be heterogeneous (i.e. few cells of the DS9 subline occupy the same region of DS8) and so better explained by more than one set of parameters. 

Figures

- Figure 2 A and C y axis 'Normalized log 2 cell count'

- Figure 3 A, Figure 4 A, these circular barplots are less readable than simpler linear ones

- Figure 3 D, E and Figure 4 D, E Axes and legends are particularly small and hard to read, please uniform them with the ones on the other panels.

Supplementary note

- Most of the supplementary note is so long, detailed and describing such basic knowledge (i.e. the description of the different types of mutations etc.) which is more appropriate for and introduction to genetics or for a PhD dissertation rather than for this manuscript.

---

## [Decision Letter · Decision Letter 2]

10 Feb 2021

Dear Dr Harris,

Thank you for submitting your revised Research Article entitled "Disentangling genetic, epigenetic, and stochastic sources of cell state variability in an in vitro model of tumor heterogeneity" for publication in PLOS Biology. I have now obtained advice from the original reviewers and have discussed their comments with the Academic Editor. 

Based on the reviews, we will probably accept this manuscript for publication, provided you satisfactorily address the remaining points raised by the reviewers. Please also make sure to address the following data and other policy-related requests.

IMPORTANT:

a) Please address the remaining concerns raised by reviewers #1 and #2.

b) Please change your title to a more declarative form. We suggest something like "An in vitro model of tumor heterogeneity resolves genetic, epigenetic, and stochastic sources of cell state variability"

c) Many thanks for depositing the code on Github. However, we will need you to make the data underlying your Figures available, either on Github or as supplementary data files (see "Data Policy" requests further down). In addition, your Data Availability Statement currently includes "Additional experimental data will be available from the corresponding author upon request" - this is not compliant with our policy, and we discourage reliance on a single named individual.

We expect to receive your revised manuscript within two weeks. 

*Published Peer Review History*

*Early Version*

Sincerely,

Roli Roberts

Senior Editor,

rroberts@plos.org,

PLOS Biology

DATA POLICY:

Regardless of the method selected, please ensure that you provide the individual numerical values that underlie the summary data displayed in the following figure panels as they are essential for readers to assess your analysis and to reproduce it: Figs 2ABCD, 3ABCDEFG, 4ABCDEFG, 5ABCDEFGH, S3ABC, S4ABCDE, S5AB, S6B, S7ABCDEFGH, S8AB, S9, S10, S11, S12, S13ABCD, S15AB, S16, S17ABC. NOTE: the numerical data provided should include all replicates AND the way in which the plotted mean and errors were derived (it should not present only the mean/average values).

REVIEWERS' COMMENTS:

Reviewer #1:

Hayford et al. PLoS Biol Manuscript #PBIOLOGY-D-20-01505R2

The revision of Hayford et al. has been considerably streamlined. The fog of readability has mostly lifted, but that has uncovered additional conceptual and technical concerns that must be addressed or clarified.

Major points

1. "Distance" on the genetic axis-epigenetic landscape (Fig. 1, 6, S14)—The authors wisely removed the evolutionary branching schematics in these cartoon figures. Now, more attention must be paid to the illustration of their conceptual paradigm. Fig. 1 schematizes the genetic axis as a linear path, one assumed to reflect the chronological accumulation of mutations. However, the vertical axis of Fig. 6 does not appear to encode any such phylogeny and instead is used a simple means to spread out the PC9 derivatives that were studied. Likewise, for the epigenetic landscape, the authors make a big deal about transitioning to adjacent basins of attraction, but what does left-vs.-right on the axis imply? Could all the cartoons be flipped or translated horizontally without any loss of meaning? Are the red-green-blue basins in Fig. 1 supposed to line up vertically or are they NOT supposed to line up? In Fig. 6, what evidence is there that DS3 is state that mutates into DS8? In Fig. S14, the implication is that DS8 derives from DS9. If the reader is not supposed to read into distance or location on the horizontal or vertical axes, what is the point of have the different PC9 derivatives share common axes at all?

2. Phylogenetic relationship between PC9-VU and PC9-MGH (Fig. S2B, S4C, S15)—The revised text gives the impression that PC9-VU and PC9-MGH had diverged from a common ancestor some unknown time ago (Fig. S2B). Lost in the weeds of the first submission was that "99% of the PC9-VU mutations were also seen in PC9-MGH" (Fig. S4C). Assuming the same genetic drift, does that not imply that PC9-VU is much closer to the ancestral line? Also, the main text should make clear that the identity of all the lines/clones was confirmed by the ~1e5 SNPs shared among PC9-MGH, PC9-VU, and PC9-BR1 (Fig. S4C); based on the language, the reader defaults to thinking that one of the lines has been misidentified. Last, it is unclear why the authors used SNPs (instead of somatic mutations not appearing as common variants in dbSNP) to evaluate the phylogenies in Fig. S15. Should not the PC9-BR1 and DS clones all derive from PC9-VU if these analyses are accurate?

3. Inappropriate use of inferCNV (Fig. 3E, 4E)—This reviewer appreciates the use of inferCNV to estimate copy number, but it is important to remind that it is only an estimate that depends on how the algorithm was deployed. Inferring gains-losses of the DS clones by using PC9-VU as the reference (Fig. 4E) is acceptable for estimating chromosomal changes relative to the parental population. Doing the same for PC9-MGH, PC9-VU, and PC9-BR1 using their average gene expression is just wrong, because all it does is map gene expression differences among the three lines to their position on chromosomes. Estimation of true gains-losses would require some type of normal reference, which would be difficult for NSCLC. A better substitute in this circumstance would be to estimate local copy number profiles directly from the copy-number estimates of the bulk whole-exome sequencing for the three.

4. Inappropriate use of UMAP (Fig. S7C, S7F)—UMAP is a fine way to reduce dimensions to preserve local and global relationships, but the dimensions should not be used to calculate subtle distances between centroids (the UMAP developer says as much: https://github.com/lmcinnes/umap/issues/92). These calculations should be removed.

5. Overinterpretation of the stochastic birth-death simulations (Fig. 5, S13)—The choice of bootstrapped Anderson-Darling recommended by Reviewer #2 is fine as a metric to compare data and stochastic simulations. However, the language of the main text is backwards that of standard hypothesis testing for frequentist statistics. A p value greater than 0.05 does not provide "strong evidence that these sublines […] are monoclonal"; only a small p value could provide evidence that the sublines are NOT monoclonal. Instead, the results suggest that the data are consistent with monoclonal population, and Occam's razor argues against invoking more complicated population structures. Compared to the results text, the methods text does a better job describing what these simulations do and do not support.

Minor points

1. Although the manuscript is much more streamlined, there remain instances of superfluous information that is unhelpful. For example, how does Fig. S1 enhance our appreciation that some genomic differences matter and others do not? Elsewhere, Fig. S12 classifies a UMAP plot based on inferred cell-cycle phase. This analysis can be done, but how does provide "further evidence that DS3, DS6, and DS7/9 correspond to three distinct cell states"? What result from the cell-cycle categorization would have falsified that there are three distinct cell states?

2. There remains some confusing duplication in the revision. In the first glance of Fig. 3D, for instance, the question arises why so many gene rows appear to have no mutations. One presumes that it is the same gene rows as shown in Fig. 4D, but the figure captions do not clarify this point. Also, although the figure is too small for row labels, there could be a handful of spaced row numbers that could be used to interpret the figure alongside the row numbers of Table S2.

Reviewer #2:

[identifies himself as Aaron S Meyer]

Hayford et al have substantially improved their manuscript by streamlining the text and figures. I appreciate the analysis provided to address my and the other reviewers' concerns. I have a couple of addressable concerns below and am supportive of publication after these minor adjustments.

Line 101: "…emerged in the absence of selective pressures." This statement could be a bit more exact. While no drug selection was applied, the cell culturing process itself or features of that environment could be providing selective pressures for continued cell line adaptation.

Lines 324-328: Since the semantic similarity relies on input about the difference between sublines, is it not differently powered to find links when there is a greater or lesser difference? Perhaps it would be better to say there is a strong genomic-transcriptomic link in DS8, but then other lines either have a weak link or insufficient changes to establish a link?

Reviewer #3:

The authors have addressed the majority of my concerns. The presentation of the results is now clearer and the manuscript is more concise, therefore I recommend its publication in Plos Biology.

---

## [Editor Report · Decision Letter 3]

4 Mar 2021

Dear Leonard,

Thank you for submitting your revised Research Article entitled "An in vitro model of tumor heterogeneity resolves genetic, epigenetic, and stochastic sources of cell state variability" for publication in PLOS Biology.

IMPORTANT: 

Sorry- we're nearly there, and we just need to finalise the data provision. Please see the Data Policy section further down. The issue is that while you point out that your experimental data are available inthe Github deposition (github.com/QuLab-VU/GES_2021), this seems to contain the raw (-ish) data and code, rather than the numerical values underlying the Figs. So...

a) Please could you add these underlying data to the Github or supply them as supplementary spreadsheets (for Figs 2ABCD, 3ABCDEFG, 4ABCDEFG, 5ABCDEFGH, S3ABC, S4ABCDE, S5AB, S6B, S7ABCDEFGH, S8AB, S9, S10, S11, S12, S13ABCD, S15AB, S16, S17ABC). Or if I'm wrong and have misunderstood the files in Github, please clarify.

b) Either way, please cite the location of the underlying data in each relevant main and supplementary Figure legend, e.g. "The data underlying this Figure can be found in github.com/QuLab-VU/GES_2021" or "The data underlying this Figure can be found in S1 Data"

We expect to receive your revised manuscript within two weeks. 

*Published Peer Review History*

*Early Version*

Sincerely,

Roli

Senior Editor,

rroberts@plos.org,

PLOS Biology

DATA POLICY:

Regardless of the method selected, please ensure that you provide the individual numerical values that underlie the summary data displayed in the following figure panels as they are essential for readers to assess your analysis and to reproduce it: Figs 2ABCD, 3ABCDEFG, 4ABCDEFG, 5ABCDEFGH, S3ABC, S4ABCDE, S5AB, S6B, S7ABCDEFGH, S8AB, S9, S10, S11, S12, S13ABCD, S15AB, S16, S17ABC. NOTE: the numerical data provided should include all replicates AND the way in which the plotted mean and errors were derived (it should not present only the mean/average values).

---

## [Editor Report · Decision Letter 4]

16 Mar 2021

Dear Leonard,

On behalf of my colleagues and the Academic Editor, Mark Siegal, I'm pleased to say that we can in principle offer to publish your Research Article "An in vitro model of tumor heterogeneity resolves genetic, epigenetic, and stochastic sources of cell state variability" in PLOS Biology, provided you address any remaining formatting and reporting issues. These will be detailed in an email that will follow this letter and that you will usually receive within 2-3 business days, during which time no action is required from you. Please note that we will not be able to formally accept your manuscript and schedule it for publication until you have made the required changes.

PRESS: We frequently collaborate with press offices. If your institution or institutions have a press office, please notify them about your upcoming paper at this point, to enable them to help maximise its impact. If the press office is planning to promote your findings, we would be grateful if they could coordinate with biologypress@plos.org. If you have not yet opted out of the early version process, we ask that you notify us immediately of any press plans so that we may do so on your behalf.

Thank you again for supporting Open Access publishing. We look forward to publishing your paper in PLOS Biology. 

Sincerely, 

Roli

Roland G Roberts, PhD 

Senior Editor 

PLOS Biology